# HAD: Hybrid Architecture Distillation for Bridging Large-Transformer Knowledge into Compact Genomic Models

## Abstract

Inspired by the great success of Masked Language Modeling (MLM) in the natural language domain, the paradigm of self-supervised pre-training and downstream fine-tuning has also achieved remarkable progress in the field of genomic sequence modeling. However, existing research often either relies on scaling up pre-training data and parameters, which brings a heavy computational burden, or lacks a systematic method to avoid the loss of prior information with compact architectures. In this work, we propose a Hybrid Architecture Distillation (HAD) approach, leveraging both distillation and reconstruction tasks for more efficient and effective pre-training. Specifically, we employ the NTv2-500M as the teacher model and devise a grouped masking strategy to align the feature embeddings of visible tokens while concurrently reconstructing the invisible tokens during MLM pre-training. To validate the effectiveness of our proposed method, we conducted comprehensive experiments on the Nucleotide Transformer Benchmark and Genomic Benchmark. Compared to models with similar parameters, our model achieved excellent performance. ***More surprisingly***, it even surpassed the *distillation ceiling*-teacher model on some sub-tasks, which is more than ***500 ×*** larger. Lastly, we conducted a comprehensive analysis of the HAD architecture, including linear probing representation evaluation, which demonstrates both the strong representation capacity of HAD and the validity of our teacher model selection for distillation. t-SNE visualization further supports these findings, providing an intuitive view of the model's representation ability.

## 1 Introduction

In the realm of DNA sequence modeling, the paradigm of self-supervised pre-training followed by fine-tuning is catalyzing significant advancements, fundamentally reshaping how genomic data is interpreted and utilized Chen et al. (2022); Zhang et al. (2024). Within this transformative landscape, masked language modeling (MLM) has prominently emerged as a primary technique. By pre-training models on vast, unlabeled DNA datasets, these methods can learn effective representations for downstream genomic tasks Ji et al. (2021); Zhou et al. (2023); Dalla-Torre et al. (2024); Nguyen et al. (2023); Schiff et al. (2024); Li et al. (2024). These learned representations are foundational for numerous downstream genomic tasks, greatly improving predictive capabilities and fostering deeper biological insights Linder et al. (2025); Li et al. (2024).

Scaling up the parameter size of models, especially in Transformers-based work Ji et al. (2021); Zhou et al. (2023); Dalla-Torre et al. (2024), is a prevalent method for enhancing pre-trained models. Though this approach can often bring certain performance gain, it inevitably results in considerably higher computational demands Ru et al. (2025).

Consequently, an alternative and also crucial research direction that focuses on novel, compact and efficient architectures has emerged Nguyen et al. (2023); Schiff et al. (2024); Consens et al. (2025). However, although these compact architectures provide desirable computational efficiency, they often struggle to match the in-depth representation learning and performance of those larger or more extensively pre-trained counterparts Dalla-Torre et al. (2024); Schiff et al. (2024). Compact models, despite their efficient pre-training capable of processing tens of billions of nucleotide tokens Schiff

et al. (2024), often hit capacity limits early, restricting their ability to learning complex patterns from massive genomic data. Conversely, larger models undergo far more extensive training Dalla-Torre et al. (2024) for deeper extraction of subtle biological feature. Thus, achieving profound representation learning in compact models remains a key challenge.

To overcome these limitations, we propose a novel framework for genomic sequence modeling with Hybrid Architecture Distillation (HAD). HAD uses a hybrid student architecture to capture a wide range of DNA sequence features, from key local feature to the global interactions, within only **1M** parameter. Based on a bidirectional Gated Delta Net (GDN) Yang et al. (2024a), it combines linear complexity with adaptive memory control via two complementary mechanisms: the gating mechanism selectively erases irrelevant or redundant non-functional sequence segments, the delta update rule accurately modifies memory by identifying specific

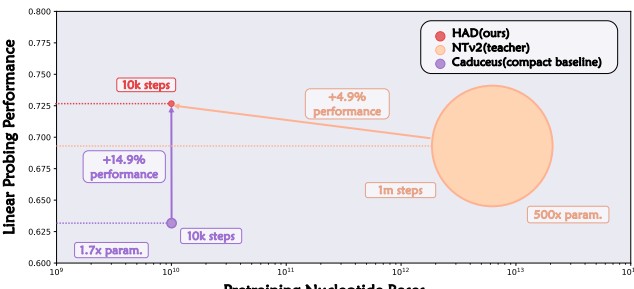

Figure 1: Comparison of model representation performance against pretraining nucleotide bases. The vertical axis shows linear probing accuracy as a measure of representation ability, as detailed in Section 4, while the horizontal axis represents pretraining data on a logarithmic scale. Circle sizes correspond to model parameters.

short sequences. To integrate comprehensive global information, this GDN backbone is augmented with a self-attention layer Dao et al. (2022); Dao (2024). This hybrid approach harnesses combined strengths, effectively integrating GDN's proficiency in capturing local and long-range sequential patterns with attention's capacity for unifying global context.

Our model empowers compact architecture with deep biological understanding through hybrid learning tasks, implemented within an innovative parallel dual-branch pretraining framework. These tasks distinctively combine two complementary objectives: a high-level feature alignment with a large teacher model using *visible* DNA nucleotides, and a low-level nucleotide reconstruction task focusing on *masked* positions. For the high-level alignment, our devised grouping masking strategy directs the student to align its feature embeddings of visible tokens with those from the teacher model, Nucleotide Transformer v2(NTv2) Dalla-Torre et al. (2024)(>500M), to gain more sophisticated biological insights. Concurrently, the low-level reconstruction branch try to predict the original identities of masked nucleotides by using the learned representation of visible nucleotides from alignment branch as context, which is implemented by a cross-attention mechanism and motivates our model to learn fundamental DNA sequence patterns and local grammar. This hybrid framework ensures HAD develops both profound representation and fine-grained understanding of DNA sequence.

To validate our proposed method's effectiveness, we conduct comprehensive evaluation on the widely used Nucleotide Transformer Benchmark and Genomic Benchmark. Our compact genomic model, with only 1M parameters, exhibits remarkable efficacy, outperforming competing models of similar size and ***surprisingly*** surpassing its large teacher model, NTv2, which has 500M parameters. Furthermore, we perform linear probing evaluation, demonstrating the robustness of HAD's representation learning and underscoring the value of NTv2 as the distillation target. More intuitive visual analysis based on t-SNE further reveals that our model effectively captures intrinsic feature patterns and discriminative genomic representations across diverse DNA categories. These results highlight the capacity of HAD to bridge the knowledge of large transformers into compact genomic models, confirming its effectiveness for both genomic sequence modeling and various downstream tasks.

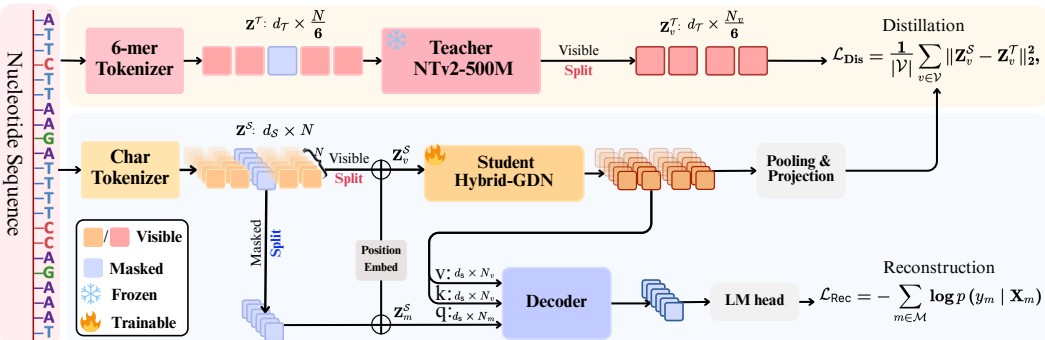

Figure 2: Proposed **H**ybrid **A**rchitecture **D**istillation (**HAD**) pre-training pipeline. The upper branch performs feature alignment on visible nucleotides, distilling *high-level* knowledge from a pre-trained teacher model to the student model. The lower branch focuses on the *low-level* reconstruction of masked nucleotides, leveraging contextual information from the student's visible nucleotide representations.

## 2 METHODOLOGY

### 2.1 OVERALL PIPELINE

Traditional self-supervised learning for DNA sequences, such as Masked Language Modeling (MLM), typically processes a partially masked input sequence $\mathbf{X}_m$ (derived from $\mathbf{X}$) through a unified encoder to predict the original nucleotides at masked positions $m$, optimized via a reconstruction objective:

$$\mathcal{L}_{\text{Rec}} = - \sum_{m \in \mathcal{M}} \log p\left(y_m \mid \mathbf{X}_m\right), \tag{1}$$

where $\mathcal{M}$ is the set of masked positions and $p\left(y_m \mid \mathbf{X}_m\right)$ represents the predicted probability of $y_i$ given $\mathbf{X}_{\mathcal{M}}$. However, conventional MLM may not fully enable compact models to learn the deep features seen in much larger, extensively pre-trained models, especially when leveraging massive datasets. Thus, our Hybrid Architecture Distillation (HAD) framework significantly innovates upon this to bridge this gap by introducing a dual-branch pipeline. This design enables synergistic learning through two distinct yet complementary objectives: high-level feature alignment on visible nucleotides and low-level reconstruction of masked nucleotides, moving beyond the single-stream processing of conventional MLM.

The overall pipeline of HAD is illustrated in Figure 2. It begins by conceptually dividing the input sequence $\mathbf{X}$ into visible nucleotides $\mathbf{X}_v$ and masked positions $\mathbf{X}_m$. In the first branch, the student model $\mathcal{S}$ processes $\mathbf{X}_v$ (via its character-level tokenizer) into a hidden representation $\mathbf{Z}_v^{\mathcal{S}}$. This is then aligned with the corresponding visible representation $\mathbf{Z}_v^{\mathcal{T}}$ derived from a large, pre-trained teacher model $\mathcal{T}$ (which processes the full $\mathbf{X}$ using its k-mer tokenizer and backbone, followed by filtering for visible parts). This feature-level distillation provides explicit high-level guidance. The second branch reconstructs the nucleotides at masked positions. For this, a decoder module integrates contextual information from the student's visible nucleotide representations $\mathbf{Z}_v^{\mathcal{S}}$ with initial embeddings derived from the masked positions $\mathbf{X}_{\mathcal{M}}$. This integration yields context-aware representations for the masked nucleotides, $\mathbf{Z}_{\mathcal{M}}^{\mathcal{S}}$. An LM Head subsequently maps these representations $\mathbf{Z}_{m\mathcal{M}}^{\mathcal{S}}$ to vocabulary logits to predict their original types, optimized with a loss analogous to $\mathcal{L}_{\text{Rec}}$. Thus, HAD distinctively conditions masked nucleotide reconstruction on information from the visible pathway, a key departure from standard MLM's reliance on local masked context alone.

### 2.2 HYBRID LEARNING TASKS

**Masking Strategy for Tokenizer Mismatch.** Traditional random nucleotide masking is unsuitable for our hybrid learning task, as such strategies can create information inconsistencies and leakage during feature alignment between the $k$-mer level teacher and character-level student models, impairing distillation. To ensure consistent information and effective feature alignment, we therefore propose a two-stage mask sampling method.

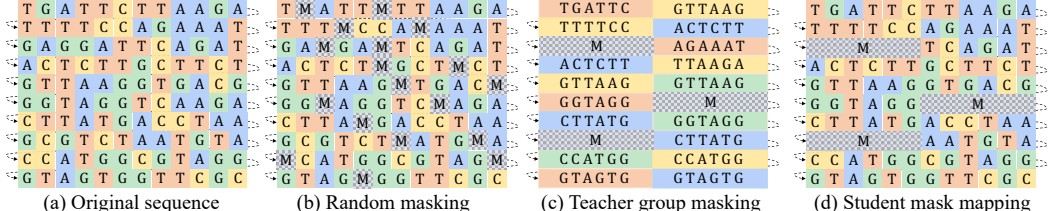

(a) Original sequence     (b) Random masking     (c) Teacher group masking     (d) Student mask mapping

Figure 3: Two-stage masking strategy in HAD. This strategy is designed to prevent information leakage and enhance feature learning for the student model during distillation.

Specifically, the first stage implements "teacher group masking" at the $k$-mer level (Figure 3c). Here, we randomly select 15% of $k$-mer units (*e.g.*, for a sequence of length $N$ and 6-mers, $N/6$ units) to define the masked regions for the teacher model. By masking entire $k$-mer blocks, it presents a more structurally coherent and challenging masked context. For our student model, which operates at a character-level, this encourages learning from larger obscured spans during distillation, thereby fostering the acquisition of more valuable, high-level features. The second stage then involves mapping these $k$-mer level mask indices to the corresponding character-level positions for the student model (Figure 3d). This two-stage approach ensures consistent mask positioning between the teacher and student, prevents information leakage, and importantly, enhances the quality of feature learning for the student through more meaningful group-level masking.

**Feature Alignment of Visible Nucleotides.** With the visible and masked nucleotide positions established by our two-stage masking strategy, the first of our hybrid learning tasks focuses on feature alignment using these visible nucleotides. These visible segments are processed through the student model's hybrid architecture (detailed in Section 2.3) to obtain its representation $\mathbf{Z}_v^{\mathcal{S}}$. This student representation is then aligned with the corresponding features $\mathbf{Z}_v^{\mathcal{T}}$ derived from a large-scale teacher model, NTv2 (a pure Transformer architecture with 500M parameters). Pre-trained on a multi-species dataset using over 1 trillion tokens, the NTv2 teacher model exhibits strong biological representation capabilities. The primary goal of this alignment is to enable the student model to inherit the teacher's sophisticated biological expertise and learned high-level features from the visible portions of the sequence.

The alignment process faces two main challenges: differing hidden dimensions and sequence length. The student model has a hidden dimension of $d_{\mathcal{S}} = 128$, while the teacher model has $d_{\mathcal{T}} = 1024$. Additionally, the student model uses a character-level tokenizer, producing sequences of length $L = N$, whereas the teacher model uses a k-mer tokenizer with $k = 6$, resulting in sequences of length $L_{\text{k-mer}} = \frac{N}{6}$. To address these, we first apply average pooling to the student model's sequence representations, reducing the sequence length from $L$ to $L_{\text{k-mer}}$ to match the teacher model's output. This is done over non-overlapping 6-mer windows. After aligning the sequence length, we use a projection layer to map the student model's hidden representations from $d_S$ to $d_T$. These two operations align both the sequence length and feature dimensions, facilitating appropriate feature alignment.

The feature alignment is achieved by minimizing the Mean Squared Error (MSE) loss between the student and teacher model representations. Since only visible nucleotides are aligned, the teacher model extracts representations for visible nucleotides based on pre-sampled mask indices. The MSE loss is computed as:

$$\mathcal{L}_{\text{Dis}} = \frac{1}{|\mathcal{V}|} \sum_{v \in \mathcal{V}} \|\mathbf{Z}_v^{\mathcal{S}} - \mathbf{Z}_v^{\mathcal{T}}\|_2^2, \tag{2}$$

where $\mathcal{V}$ denotes the set of visible nucleotide positions. Within the sum, $\mathbf{Z}_v^{\mathcal{S}}$ is the student model's representation at a visible position $v \in \mathcal{V}$, and $\mathbf{Z}_v^{\mathcal{T}}$ is the teacher model's representation at the same position $v$. By minimizing this loss, we ensure that the student model effectively aligns its visible nucleotide representations with the teacher model's.

**Reconstruction of Masked Nucleotides.** To preserve the model's capability for low-level nucleotide understanding, a masked nucleotide reconstruction task remains essential. However, due to our framework's clear division of visible and masked nucleotide processing pathways, a dedicated decoder mechanism is necessary for this reconstruction. In our approach for this task, the masked

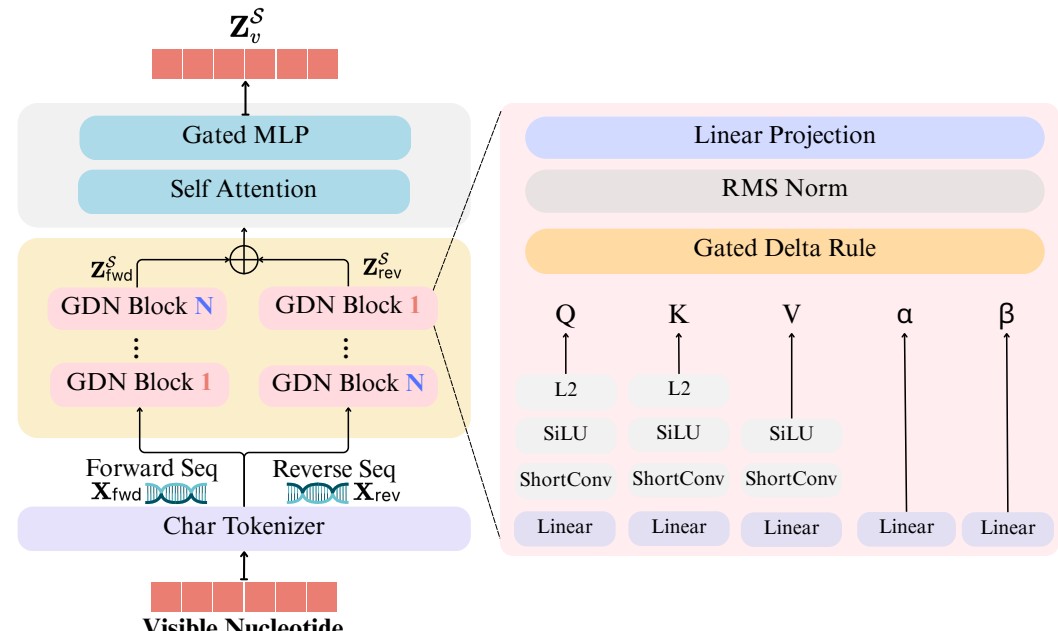

Figure 4: Hybrid architecture of our student model, combining a bidirectional Gated Delta Net (GDN) backbone with a self-attention layer for efficient sequential processing and global context integration within a compact 1.1M parameter budget. Within the GDN, $\alpha$ serves as a data-dependent gate controlling memory erasure, while $\beta$ acts as the update strength from the delta rule.

nucleotide positions are initialized randomly rather than using a fixed [MASK] token. We also ensure that positional information is added to the representations of both visible and masked nucleotides before they enter their respective model pathways. This explicit positional encoding preserves crucial spatial information, enabling accurate reconstruction of the original masked nucleotide positions.

For the Decoder, we use Cross Attention(CA), where the masked nucleotides act as the query, and the visible nucleotides serve as both the key and value. The masked nucleotides' representations, as queries, attend to the visible nucleotides' representations to produce the corresponding masked sequence. The CA operation is computed as follows:

$$\text{CA}\left(\mathbf{Q}_m, \mathbf{K}_v, \mathbf{V}_v\right) = \text{softmax}\left(\frac{\mathbf{Q}_m \mathbf{K}_v^T}{\sqrt{d_k}}\right)\mathbf{V}_v, \tag{3}$$

where $\mathbf{Q}_m$ is the query (masked nucleotides), $\mathbf{K}_v$ and $\mathbf{V}_v$ are the key and value (visible nucleotides), and $d_k$ is the dimensionality of the keys. This attention mechanism allows the model to focus on the relevant information in the visible nucleotides while reconstructing the masked nucleotides.

Finally, an LM Head maps the resulting masked representations $\mathbf{Z}_m^S$ to vocabulary logits, producing a probability distribution for each masked nucleotide. The optimization objective for this reconstruction is to minimize a cross-entropy(CE) function analogous to $\mathcal{L}_{\text{Rec}}$ (Same as equation 1).

## 2.3 HYBRID STUDENT ARCHITECTURE

Our hybrid architecture for DNA sequence modeling is founded on the Gated Delta Net (GDN) Yang et al. (2024b;a). The state $\mathbf{S}_t$ of GDN at each time step $t$ is dynamically updated based on its previous state $\mathbf{S}_{t-1}$ and current inputs, following the principle:

$$\mathbf{S}_t = \mathbf{S}_{t-1}\left(\alpha_t\left(\mathbf{I} - \beta_t \boldsymbol{k}_t \boldsymbol{k}_t^T\right)\right) + \beta_t \boldsymbol{v}_t \boldsymbol{k}_t^T. \tag{4}$$

Equation equation 4 details how GDN dynamically updates its memory $\mathbf{S}_t$ for our DNA sequence modeling, using complementary "Gated" and "Delta Rule" mechanisms. The gating, via $\alpha_t$ and the $\left(\mathbf{I} - \beta_t \boldsymbol{k}_t \boldsymbol{k}_t^T\right)$ term, allows the model to selectively clear or retain prior nucleotide context, effectively filtering information from less relevant DNA segments based on the current key $\boldsymbol{k}_t$. The Delta Rule

Table 2: **Nucleotide Transformer Benchmark Results.** Performance of HAD against baseline models and its NTv2 teacher under fine-tuning evaluation on each downstream task. Results are means from 10-fold cross-validation with 10 random seeds; best performance is in **bold**, second-best is underlined. Error bars represent the range (maximum-minimum) across the 10 seeds. The final column, $\Delta_{\text{Student-Teacher}}$, shows the performance difference between HAD and its teacher.

| Dataset
Param. | Enformer
252M | DNABERT-2
117M | HyenaDNA
1.6M | Caduceus
1.9M | NT(Teacher)
498.3M | HAD(Student)
1.1M | $\Delta_{\text{Student-Teacher}}$
-497.2M |
|---|---|---|---|---|---|---|---|
| ***Histone Markers*** | | | | | | | |
| H3 | 0.719 ±0.048 | 0.785 ±0.033 | 0.779 ±0.037 | 0.815 ±0.048 | 0.784 ±0.047 | **0.822** ±0.007 | +3.8% |
| H3K14ac | 0.288 ±0.077 | 0.591 ±0.028 | 0.612 ±0.065 | 0.631 ±0.026 | 0.551 ±0.021 | **0.684** ±0.041 | +13.6% |
| H3K36me3 | 0.344 ±0.055 | 0.591 ±0.020 | 0.613 ±0.041 | 0.601 ±0.129 | 0.625 ±0.013 | **0.653** ±0.031 | +2.8% |
| H3K4me1 | 0.291 ±0.061 | 0.511 ±0.028 | 0.512 ±0.024 | 0.523 ±0.039 | 0.550 ±0.021 | **0.571** ±0.037 | +2.1% |
| H3K4me2 | 0.211 ±0.069 | 0.336 ±0.040 | 0.455 ±0.095 | 0.487 ±0.170 | 0.319 ±0.045 | **0.562** ±0.033 | +24.3% |
| H3K4me3 | 0.158 ±0.072 | 0.352 ±0.077 | 0.549 ±0.056 | 0.544 ±0.045 | 0.410 ±0.033 | **0.643** ±0.052 | +23.3% |
| H3K79me3 | 0.496 ±0.042 | 0.613 ±0.030 | 0.672 ±0.048 | 0.697 ±0.077 | 0.626 ±0.026 | **0.712** ±0.022 | +8.6% |
| H3K9ac | 0.420 ±0.063 | 0.542 ±0.029 | 0.581 ±0.061 | 0.622 ±0.030 | 0.562 ±0.040 | **0.656** ±0.023 | +9.4% |
| H4 | 0.732 ±0.076 | 0.796 ±0.027 | 0.763 ±0.044 | **0.811** ±0.022 | 0.799 ±0.025 | 0.806 ±0.018 | +0.7% |
| H4ac | 0.273 ±0.063 | 0.463 ±0.041 | 0.564 ±0.038 | 0.621 ±0.054 | 0.495 ±0.032 | **0.654** ±0.039 | +15.9% |
| ***Enhancer Annotation*** | | | | | | | |
| Enhancer | 0.451 ±0.108 | 0.516 ±0.098 | 0.517 ±0.117 | 0.546 ±0.073 | 0.548 ±0.144 | **0.571** ±0.075 | +2.3% |
| Types | 0.309 ±0.134 | 0.423 ±0.051 | 0.386 ±0.185 | 0.439 ±0.054 | 0.424 ±0.132 | **0.467** ±0.073 | +4.3% |
| ***Promoter Annotation*** | | | | | | | |
| All | 0.954 ±0.006 | 0.971 ±0.006 | 0.960 ±0.005 | 0.970 ±0.004 | **0.976** ±0.006 | 0.968 ±0.004 | -0.8% |
| Non-TATA | 0.955 ±0.010 | 0.972 ±0.005 | 0.959 ±0.008 | 0.969 ±0.011 | **0.976** ±0.005 | 0.968 ±0.005 | -0.8% |
| TATA | 0.960 ±0.023 | 0.955 ±0.021 | 0.944 ±0.040 | 0.953 ±0.016 | **0.966** ±0.013 | 0.958 ±0.009 | -0.8% |
| ***Splice Site Annotation*** | | | | | | | |
| All | 0.848 ±0.019 | 0.939 ±0.009 | 0.956 ±0.011 | 0.940 ±0.027 | **0.983** ±0.008 | 0.911 ±0.016 | -7.2% |
| Acceptor | 0.914 ±0.028 | 0.975 ±0.006 | 0.958 ±0.010 | 0.937 ±0.033 | **0.981** ±0.011 | 0.858 ±0.016 | -12.3% |
| Donor | 0.906 ±0.027 | 0.963 ±0.006 | 0.949 ±0.024 | 0.948 ±0.025 | **0.985** ±0.022 | 0.887 ±0.045 | -9.8% |

component, $\beta_t \boldsymbol{v}_t \boldsymbol{k}_t^T$, then precisely incorporates features from the current input nucleotides (key $\boldsymbol{k}_t$, value $\boldsymbol{v}_t$), ensuring significant DNA patterns update the memory. To enable bidirectional modeling, we process the original sequence $\mathbf{X}_{\text{fwd}}$ and its reverse $\mathbf{X}_{\text{rev}}$ with separate GDN modules, yielding forward $\mathbf{Z}_{\text{fwd}}^{\mathcal{S}}$ and reverse $\mathbf{Z}_{\text{rev}}^{\mathcal{S}}$ hidden states. These are combined by reversing $\mathbf{Z}_{\text{rev}}^{\mathcal{S}}$ and adding it to $\mathbf{Z}_{\text{fwd}}^{\mathcal{S}}$, allowing the model to capture dependencies from both upstream and downstream contexts. To achieve efficient GPU utilization, GDN is parallelized using a chunk-wise method.

To further integrate global sequence information, the bidirectional GDN backbone is enhanced with an attention mechanismVaswani et al. (2017). Specifically, following the final bidirectional GDN module, we append a single self-attention layer, implemented using Flash Attention Dao et al. (2022); Dao (2024). The output from this attention layer is then processed

Table 1: Comparison of parameter sizes and GFLOPs across models.

| Model | NTv2(500M) | Caduceus | HAD |
|---|---|---|---|
| Core Arch. | Transformer | Bi-Mamba | Hybrid Bi-GDN |
| Param. | 500M | 1.9M | 1.1M |
| GFLOPs (L=512) | 326.95 | 3.39 | 1.46 |
| GFLOPs (L=1026) | 686.53 | 6.79 | 3.05 |

by a simple Gated MLP. This completes our hybrid architecture, designed to balance efficient sequential modeling with global contextual understanding. A quantitative comparison of parameter sizes and computational costs (GFLOPs) with baseline models is summarized in Table 1.

# 3 EXPERIMENTS

In this section, we present the experimental setup and results for evaluating our proposed method. Our model is evaluated against state-of-the-art baselines on the Nucleotide Transformer and Genomic BenchmarksGrešová et al. (2023). We also use t-SNE for visualization, aiming to validate the feature learning capability transferred from the teacher to the student model within the HAD framework.

## 3.1 EXPERIMENTS SETTING

**Pre-training.** We employed the hybrid architecture that incorporates both distillation and reconstruction tasks as described in Section 2. For comparison with baseline models, we used the exact same pre-training dataSchneider et al. (2017) as in Nguyen et al. (2023); Schiff et al. (2024), which adopts the training/validation split proposed by Avsec et al. (2021). When pretraining on the human

Table 3: **Genomic Benchmarks Results.** Performance of HAD against baseline models under fine-tuning evaluation. Results are means from 5-fold cross-validation with 5 random seeds. The best performance in each row is in **bold**, and the second-best is underlined. Error bars represent the range (maximum-minimum) across the random seeds. The final row shows the average performance across all eight tasks, demonstrating HAD's strong overall results on this benchmark.

| Dataset | CNN
Grešová et al. (2023) | HyenaDNA
Nguyen et al. (2023) | Mamba
Schiff et al. (2024) | Caduceus
Schiff et al. (2024) | HAD |
|---|---|---|---|---|---|
| Mouse Enhancers | 0.715 ±0.087 | 0.780 ±0.025 | 0.743 ±0.054 | 0.754 ±0.074 | **0.788** ±0.033 |
| Coding vs Intergenomic | 0.892 ±0.008 | 0.904 ±0.005 | 0.904 ±0.004 | **0.915** ±0.003 | 0.913 ±0.003 |
| Human vs Worm | 0.942 ±0.002 | 0.964 ±0.002 | 0.967 ±0.002 | **0.973** ±0.001 | 0.971 ±0.001 |
| Human Enhancer Cohn | 0.702 ±0.021 | 0.729 ±0.014 | 0.732 ±0.029 | **0.747** ±0.004 | 0.744 ±0.010 |
| Human Enhancer Ensembl | 0.744 ±0.122 | 0.849 ±0.006 | 0.862 ±0.008 | 0.893 ±0.008 | **0.909** ±0.004 |
| Human Regulatory | 0.872 ±0.005 | 0.869 ±0.012 | 0.814 ±0.211 | 0.872 ±0.011 | **0.882** ±0.012 |
| Human OCR Ensembl | 0.698 ±0.013 | 0.783 ±0.007 | 0.815 ±0.002 | 0.828 ±0.006 | **0.832** ±0.003 |
| Human NonTATA Promoters | 0.861 ±0.009 | 0.944 ±0.002 | 0.933 ±0.007 | 0.946 ±0.007 | **0.960** ±0.008 |
| **Average** | 0.803 | 0.853 | 0.846 | 0.866 | **0.875** |

reference genomeSchneider et al. (2017), we followed the RC equivariance inductive bias proposed by Schiff et al. (2024), implementing it using data augmentation, which has been proven to be an effective and straightforward approach. We chose a sequence length of 1026 for two key reasons: it's suitable for our downstream tasks (most sequences in Nucleotide Transformer Benchmarks and Genomic benchmarks are $< 1k$ bp), and its divisibility by 6 (the teacher's k-mer size) helps resolve tokenizer mismatches between our character-level student model and the $k$-mer based teacher model. Our student model itself is configured with 4 Gated Delta Net (GDN) blocks, each with a dimension of 128, resulting in a compact model with approximately 1.1 million parameters. Regarding the teacher model, NTv2-500M was selected as the source of high-level knowledge, providing rich feature representations.

**Fine-tuning.** We performed supervised training for each downstream task in both the Nucleotide Transformer benchmarks and the Genomic Benchmarks. Our fine-tuning protocol, including the use of post-hoc conjoiningZhou et al. (2022a) for model RC invariance, strictly followed the configurations outlined in Schiff et al. (2024). To ensure a fair comparison, all baselines and their reported results were adopted directly from Schiff et al. (2024), reflecting our identical experimental setup. Evaluation metrics were chosen per benchmark: for Nucleotide Transformer Benchmarks, following Nguyen et al. (2023); Schiff et al. (2024), we used Matthews Correlation Coefficient (MCC) for histone marker tasks, F1 score for enhancer, promoter, and splice site annotation tasks (with accuracy for the "splice site all" task). All Genomic Benchmark tasks were evaluated using Top-1 accuracy.

## 3.2 DOWNSTREAM EVALUATION

**Nucleotide Transformer Benchmarks.** The evaluation of our proposed HAD model on the Nucleotide Transformer Benchmarks is presented in Table 2. With only 1.1M parameters, HAD is the most compact model among all baselines, yet it demonstrates exceptional performance. It achieves leading results in the majority of Histone Marker tasks and all Enhancer Annotation tasks, securing the top position in 11 out of 18 tasks overall. Notably, as highlighted in the $\Delta_{\text{Student-Teacher}}$ column, HAD consistently outperforms its significantly larger teacher model (NTv2, 500M parameters) across numerous tasks, despite utilizing approximately 497.2M fewer parameters. This outcome underscores that the feature alignment process integral to HAD not only facilitates effective knowledge transfer but also empowers the student model to surpass the teacher's performance ceiling, thereby significantly enhancing its learning and capabilities on downstream genomic tasks.

**Genomic Benchmarks.** We further evaluated HAD on the Genomic Benchmarks, with results detailed in Table 3. The selection of baseline models for comparison remained consistent with those in Schiff et al. (2024). In five of the eight downstream tasks within these benchmarks, our HAD model achieved the highest score among the evaluated baselines. Encouragingly, HAD's average performance score of **0.875** across all tasks in these benchmarks was the highest among all reported baselines.

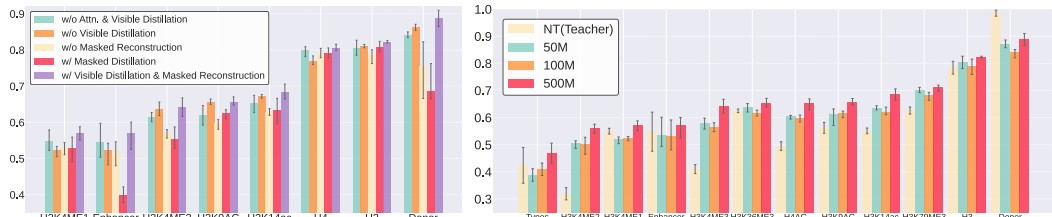

Figure 5: Ablation on pretraining scheme with different model architectures (left) and teacher model size (right)

Table 4: Validation loss for different architectures.

| Arch. | Step | | | | |
|---|---|---|---|---|---|
| | 2k | 4k | 6k | 8k | 10k |
| Masked Language Modeling | | | | | |
| GDN w/o Attn. | 1.0422 | 1.0290 | 1.0202 | 1.0127 | 1.0090 |
| GDN w/ Attn. | 1.0489 | 1.0273 | 1.0154 | 1.0068 | 1.0033 |
| Masked Language Modeling + Distillation | | | | | |
| $\mathcal{M}$ Dis. | 0.3167 | 0.3149 | 0.3111 | 0.2994 | 0.2901 |
| $\mathcal{V}$ Dis. & $\mathcal{M}$ Rec. | 0.3177 | 0.3164 | 0.3143 | 0.3118 | 0.3049 |

Table 5: Perplexity scores with different teacher models.

| Teacher Model | Step | | | | |
|---|---|---|---|---|---|
| | 2k | 4k | 6k | 8k | 10k |
| Masked Language Modeling + Distillation | | | | | |
| 50M | 5.876 | 5.835 | 5.814 | 5.797 | 5.785 |
| 100M | 5.720 | 5.679 | 5.646 | 5.622 | 5.609 |
| 500M | 5.217 | 4.903 | 4.708 | 4.599 | 4.538 |

## 3.3 Ablation Study

**Ablation of Different Architectures.** To investigate the effectiveness of our overall HAD framework, we conducted an ablation study on the Nucleotide Transformer benchmarks (Figure 5 left). Our full model ("HAD w/ Visible Distillation & Masked Reconstruction"), with its hybrid architecture and dual-branch pretraining (visible distillation and masked reconstruction), was compared against variants lacking key components or using altered training strategies. These included removing self-attention and visible distillation ("HAD w/o Attn. & Visible Distillation"; GDN backbone, MLM-only), omitting only visible distillation ("HAD w/o Visible Distillation"; hybrid architecture, MLM-only), and distilling only from masked positions ("HAD w/ Masked Distillation"). The full HAD model significantly outperformed these ablations, highlighting that its hybrid architecture and dual-branch learning strategy are crucial for its superior performance.

Table 4 further details the pre-training losses for these ablated architectures. Among the MLM-exclusive versions ("GDN w/o Attn." and "GDN w/ Attn."), both targeted masked nucleotide reconstruction; the attention-equipped version achieved lower CE reconstruction loss, indicating more effective pre-training. For distillation approaches, the MSE loss from "$\mathcal{M}$ Dis." (masked distillation) was lower than the visible distillation MSE component of "$\mathcal{V}$ Dis. & $\mathcal{M}$ Rec.". This discrepancy with the latter model's established superior downstream performance could be explained by two factors: first, visible distillation is more challenging due to a larger number of targets; and second, component pre-training losses often misalign with overall downstream task performance.

**Ablation of Different Teacher Models.** We further investigated the impact of teacher model size on distillation performance by ablating the teacher component, employing NTv2 variants with 50M, 100M, and 500M parameters. As illustrated in Figure 5 (right), a significant enhancement in the student model's downstream task performance was contingent upon guidance from the 500M parameter teacher. This observation highlights that a teacher model must possess substantial representational capacity, likely a product of comprehensive pretraining on extensive nucleotide data, to serve as an effective source of rich features for successful distillation to a smaller student architecture.

Further supporting this, the pre-training perplexity scores of these teacher models (Table 5) show a clear hierarchy: the 500M teacher achieved the lowest perplexity, followed by the 100M and 50M models across all evaluated training steps. This superior intrinsic language modeling capability of the largest teacher model likely underpins its effectiveness as a richer feature source for distillation.

## 4 Representation Analysis

### 4.1 Linear Probing Analysis

Downstream task performance alone does not capture the full extent of a model's representational ability. To assess this, we evaluate models based on their few-shot representation ability, which measures their ability to generalize to unseen biological data without fine-tuning.

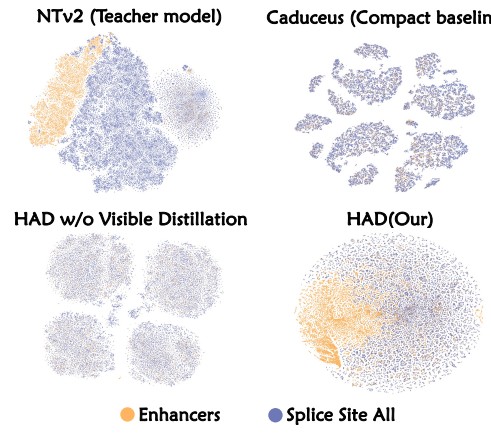

Figure 6: t-SNE visualization of pre-trained model representations, highlighting HAD's effective knowledge transfer from NTv2 through the distillation branch, particularly for distinguishing enhancer-related features.

| Dataset | Caduceus | NTv2 | HAD |
|---|---|---|---|
| H3K14ac | 0.615 | **0.710** | 0.650 |
| H3K36me3 | 0.615 | **0.839** | 0.650 |
| H3K4me1 | 0.615 | 0.677 | **0.688** |
| H3K4me2 | 0.539 | **0.677** | 0.625 |
| H3K4me3 | 0.539 | **0.710** | 0.613 |
| H3K79me3 | 0.615 | 0.613 | **0.713** |
| Promoter | 0.769 | 0.615 | **0.920** |
| Splice Acceptors | 0.692 | 0.692 | **0.812** |
| Splice Donors | 0.692 | 0.692 | **0.863** |
| **Mean** | 0.632 | 0.692 | **0.726** |

Figure 7: Linear Probing Representation Performance comparison of Caduceus, NTv2, and HAD on NT datasets. Each column shows Test Accuracy (Acc) and Test AUC.

We use a linear probe on the frozen hidden states of each model, trained on genomic sequences from Nucleotide Transformer tasks. The final hidden states are extracted from the pre-trained models, and a linear classifier is trained on these features for binary classification. This setup evaluates the model's few-shot ability to generalize to unseen biological data without fine-tuning.

Figure 7 shows that HAD outperforms both NTv2 and Caduceus in few-shot representation ability, demonstrating its superior generalization across a range of biological tasks. HAD builds on the representations learned by NTv2, refining them to better capture biologically meaningful patterns while maintaining the model's compactness. This result highlights that HAD, through its design and integration of NTv2's representations, is not only effective in downstream tasks but also excels in generalization, further establishing its strength as a compact yet highly capable model.

## 4.2 QUALITATIVE ANALYSIS

To investigate the effectiveness of feature learning specifically imparted by the distillation process within our HAD framework, we employed t-SNE Van der Maaten & Hinton (2008) to visualize representations from several models. We compared our full HAD model against the teacher model (NTv2-500M Dalla-Torre et al. (2024)), Caduceus Schiff et al. (2024), and the ablation variant "HAD w/o Visible Distillation" described in Section 3.3. These models were used as feature extractors for downstream task data from *Enhancers* and *splice site all*. As visualized in Figure 6, the NTv2-500M teacher model exhibits a strong capability to distinguish *Enhancers*. In contrast, both Caduceus and the "HAD w/o Visible Distillation" ablation fail to form meaningful clusters for these enhancer features. Conversely, our complete HAD model clearly learns and separates these enhancer-related features, effectively mirroring its teacher's discriminative ability. This t-SNE analysis underscores that HAD successfully acquires high-level feature representations from the teacher model through the proposed distillation mechanism.

## 5 CONCLUSION

We have introduced HAD, an effective framework for DNA sequence modeling that utilize hybrid learning tasks, integrating feature alignment with masked nucleotide reconstruction. Our compact (1.1M parameter) student model thereby learns rich, high-level biological features by distilling knowledge from an extensively pre-trained and significantly larger teacher. Across different downstream tasks, HAD not only outperformed most models with comparable parameters but also surprisingly exceeded the performance of its $500\times$ larger teacher model. Finally, both quantitative linear probing and qualitative t-SNE analyses confirmed HAD's effective knowledge transfer, balancing compactness with robust discriminative feature learning.

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

APPENDICES

# A  RELATED WORK

## A.1  NETWORK ARCHITECTURE FOR DNA MODELING

In recent years, research in DNA sequence modeling has increasingly focused on the development of more efficient model architectures, particularly in the context of Transformer-based models. Notable works such as DNABERT Ji et al. (2021), DNABERT2 Zhou et al. (2023), and Nucleotide Transformer Dalla-Torre et al. (2024) have successfully employed standard Transformers as their backbone networks, achieving impressive performance in genomic sequence tasks. However, these models are not without limitations, especially their scalability to long-sequence modeling and their relatively high inference costs. To address these challenges, recent advancements have turned to more efficient modeling approachesFan et al. (2024d;c; 2023; 2024b;a); Gu & Dao (2023); Dao & Gu (2024); Yang et al. (2024b;a); Behrouz et al. (2024); Sun et al. (2023). For instance, HyenaDNA Nguyen et al. (2023) introduces the Hyena operator, which reduces the model size to approximately 6.6M parameters while extending the model's capacity to handle sequences up to 1M in length. Similarly, Caduceus Schiff et al. (2024) proposes a bidirectional and RC-equivariant Mamba block as the backbone, successfully incorporating the concept of selective structured state space models (SSMs)Gu & Dao (2023); Dao & Gu (2024) into the domain of DNA sequence modeling. In the realm of RNN-based models, recent studies have enhanced the global modeling capacity by incorporating a limited number of attention layers into the architecture. Additionally, recent works have introduced novel computational strategies, such as the Delta Rule Yang et al. (2024b;a) and the Titans Behrouz et al. (2024), which aim to improve memory management and retrieval performance for sequence modeling tasksArora et al. (2023); Wen et al. (2024); Yin et al. (2025); Akyürek et al. (2024). These developments indicate that adopting more efficient architectural designs and advanced computational strategies can overcome the inherent limitations of existing models, offering promising avenues for progress in DNA sequence modeling. This paper explores the potential of hybrid architectures as backbone networks for DNA modeling, aiming to advance the field through these innovative approaches.

## A.2  KNOWLEDGE DISTILLATION FOR DNA MODELING

Knowledge distillation (KD) Gou et al. (2021); Tang et al. (2020); Busbridge et al. (2025) is a model compression technique that transfers knowledge from a large, high-capacity teacher model to a lightweight student model, enabling the latter to mimic the teacher's behavior while reducing computational costs. Initially proposed by Hinton et al. (2015), KD leverages soft targets derived from the teacher's output distribution rather than relying solely on ground-truth labels, thereby capturing richer inter-class relationships and enhancing generalization. Over time, KD has evolved into diverse paradigms, including feature-based distillation (*e.g.*, aligning intermediate representations), contrastive distillation (*e.g.*, preserving sample similarity structures), and relational distillation (*e.g.*, modeling geometric relationships). Recent advancements extend KD to cross-architecture settings, enabling knowledge transfer between heterogeneous model families (*e.g.*, Transformer→MLP), and self-distillation frameworks where the student iteratively refines its own outputs. In masked image modeling (MIM)Cao et al. (2020); Dong et al. (2023); Xie et al. (2022); Woo et al. (2023); Oquab et al. (2023); Zhou et al. (2022b), where models learn by reconstructing masked regions of images, distillation has been instrumental in compressing large vision transformers (ViTs). For instance, feature-based distillation aligns intermediate attention maps between teacher and student models, preserving spatial-semantic patterns critical for reconstruction. While KD in DNA pretraining remains underexplored, insights from related domains suggest promising directions, such as Distilled DeepConsensus Belyaeva et al. (2022) and FinDNA Yu et al. (2025), which applied KD and self-distillation techniques in the DNA correction and prediction tasks respectively. Analogously, DNA pretraining could leverage feature distillation to align latent representations of genomic sequences between teacher and student models, preserving motifs and regulatory patterns. In this paper, we will dive into this question and explore the potential of the knowledge distillation for hybrid architectures.

## B    LIMITATIONS AND FUTURE WORK

Our proposed method delivers remarkable performance across a range of genomic and nucleotide downstream tasks. However, it has a notable limitation in that it necessitates knowledge distillation from a teacher model that is nearly 500 million parameters in size. This teacher model is significantly larger than our model and demands substantial memory resources. Consequently, the development of a more efficient distillation paradigm for DNA modeling emerges as a pressing need. In addition to addressing this limitation, we also recognize the significant value of modeling long DNA sequences. Long sequence DNA modeling has the potential to unlock deeper insights into genomic information and enable more advanced applications. Another limitation lies in the distillation alignment approach, which currently lacks further exploration. This HAD architecture with high-level alignment and mask reconstruction techniques we use represent an initial attempt, and future work could benefit from incorporating more advanced distillation frameworks from other fields to further enhance performance.

## C    METHOD DETAILS

### C.1    CHUNKWISE PARALLEL GATED DELTA NET

The Gated Delta Net (GDN), introduced in Section 2.3 and defined by the sequential update rule given in Equation equation 4, presents inherent challenges for efficient parallelization on modern GPUs due to its recurrent nature. This can lead to hardware under-utilization and limited throughput, particularly when processing long DNA sequences. While some linear recurrences can be fully parallelized using scan operations across the entire sequence length, direct application of such techniques to DeltaNet-like architectures often incurs prohibitive computational costs (e.g., potentially $O(L \log L d^3)$ FLOPs for sequence length $L$ and dimension $d$) and substantial memory requirements for intermediate states ($O(Ld^2)$) Yang et al. (2024b). To overcome these limitations and ensure hardware-efficient training, our student model employs the chunkwise parallel computation method for the GDN backbone, as detailed in Yang et al. (2024b;a). This technique processes the input sequence in smaller, manageable chunks, enabling substantial parallel execution within each chunk while limiting inter-chunk state materialization. The chunkwise method can introduce a marginal increase in the total number of arithmetic operations compared to an idealized sequential recurrent process, yet it generally maintains the same asymptotic dependence on sequence length. Crucially, these potential minor costs are outweighed by the significant practical advantages of enhanced hardware utilization and optimized GPU memory access, resulting in considerably faster training times.

The core of the chunkwise parallel form lies in expressing the state update over a segment (or chunk) of $r$ steps in a manner amenable to parallel computation. By partially unrolling the recurrence relation (Equation equation 4), the state $\mathbf{S}_{[t]}^r$ at the end of a chunk of $r$ steps, originating from an initial state $\mathbf{S}_{[t]}$ for that chunk, can be written as:

$$\mathbf{S}_{[t]}^r = \mathbf{S}_{[t]} \underbrace{\left( \prod_{i=1}^{r} \alpha_{[t]}^i \left( \mathbf{I} - \beta_{[t]}^i \boldsymbol{k}_{[t]}^i \boldsymbol{k}_{[t]}^{i\top} \right) \right)}_{:=\mathbf{P}_{[t]}^r} + \underbrace{\sum_{i=1}^{r} \left( \beta_{[t]}^i \boldsymbol{v}_{[t]}^i \boldsymbol{k}_{[t]}^{i\top} \prod_{j=i+1}^{r} \alpha_{[t]}^j \left( \mathbf{I} - \beta_{[t]}^j \boldsymbol{k}_{[t]}^j \boldsymbol{k}_{[t]}^{j\top} \right) \right)}_{:=\mathbf{H}_{[t]}^r} \quad (5)$$

In this formulation, $\mathbf{P}_{[t]}^r$ represents the cumulative linear transformation applied to the initial state $\mathbf{S}_{[t]}$ over the $r$ steps within the chunk, and $\mathbf{H}_{[t]}^r$ accumulates the contributions from the inputs $\boldsymbol{v}_{[t]}^i$ (associated with keys $\boldsymbol{k}_{[t]}^i$) within the chunk. As demonstrated in Yang et al. (2024b;a), the terms $\mathbf{P}_{[t]}^r$ and $\mathbf{H}_{[t]}^r$ can be computed efficiently in parallel using the following expressions:

$$\mathbf{P}_{[t]}^r = \gamma_{[t]}^r \left( \mathbf{I} - \sum_{i=1}^{r} \mathbf{w}_{[t]}^i \boldsymbol{k}_{[t]}^{i\top} \right) \quad , \quad \mathbf{H}_{[t]}^r = \sum_{i=1}^{r} \frac{\gamma_{[t]}^r}{\gamma_{[t]}^i} \mathbf{u}_{[t]}^i \boldsymbol{k}_{[t]}^{i\top} \quad (6)$$

where $\gamma_{[t]}^k = \prod_{j=1}^{k} \alpha_{[t]}^j$. The auxiliary vectors $\mathbf{w}_{[t]}^i$ and $\mathbf{u}_{[t]}^i$ (for $i = 1, \ldots, r$ within the chunk) are computed using the following recurrences. These recurrences can themselves be structured for

efficient parallel computation within the chunk, typically via parallel scan algorithms:

$$\mathbf{w}_{[t]}^i = \beta_{[t]}^i \left( \boldsymbol{k}_{[t]}^i - \sum_{j=1}^{i-1} \left( \mathbf{w}_{[t]}^j \left( \boldsymbol{k}_{[t]}^{j\top} \boldsymbol{k}_{[t]}^i \right) \right) \right) \tag{7}$$

$$\mathbf{u}_{[t]}^i = \beta_{[t]}^i \left( \boldsymbol{v}_{[t]}^i - \sum_{j=1}^{i-1} \left( \mathbf{u}_{[t]}^j \left( \frac{\gamma_{[t]}^i}{\gamma_{[t]}^j} \boldsymbol{k}_{[t]}^{j\top} \boldsymbol{k}_{[t]}^i \right) \right) \right) \tag{8}$$

It is also common practice in the implementation of such architectures Yang et al. (2024a) to employ normalization techniques (e.g., L2 normalization of keys and queries) to enhance training stability, although these are not explicitly shown in the core update rules above. This chunkwise parallelization mechanism is instrumental in allowing the GDN backbone of our student model to be trained efficiently on modern GPU hardware, effectively harnessing their parallel computation capabilities for processing DNA sequences encountered in genomic research.

## C.2 REPRESENTATION ALIGNMENT BETWEEN STUDENT AND TEACHER

---
**Algorithm 1** Average Pooling and Final Projection Operation

---
1: **Input:** Sequence $X \in \mathbb{R}^{B \times L_{in} \times D_S}$; Pooling factor $P_F$.
2: **Module Layers (initialized components):**
3:     *Pooling Block:*
4:         $\text{Linear}_{\text{pool}} : \mathbb{R}^{D_S} \to \mathbb{R}^{D_S}$ (Internal linear projection)
5:         LayerNorm: Layer normalization module
6:         Dropout: Dropout module
7:     *Projection Layer:*
8:         $\text{Projection} : \mathbb{R}^{D_S} \to \mathbb{R}^{D_T}$ (Projects to teacher dimension)
9: **Output:** Projected sequence $X_{\text{final}} \in \mathbb{R}^{B \times L_{out} \times D_T}$, where $L_{out} = \lceil L_{in}/P_F \rceil$.

10: **if** $L \pmod{P_F} \neq 0$ **then**
11:     $L_{padded} \leftarrow \lceil L/P_F \rceil \times P_F$
12:     $X \leftarrow X$ padded to length $L_{padded}$ along the sequence dimension
13: **end if**
14: $L_{out} \leftarrow L/P_F$
15: $X_{\text{grouped}} \leftarrow$ Reshape $X$ from $(B, L_{padded}, D_S)$ into $(B, L_{out}, P_F, D_S)$
16: $X_{\text{mean}} \leftarrow$ Mean of $X_{\text{grouped}}$ along the dimension of $P_F$
17: $X_{\text{pool\_out}} \leftarrow \text{Dropout}(\text{LayerNorm}(\text{Linear}_{\text{pool}}(X_{\text{mean}})))$
18: $X_{\text{final}} \leftarrow \text{Projection}(X_{\text{pool\_out}})$
19: **return** $X_{\text{final}}$

---

The main text (Section 2) explains why aligning student and teacher representations is necessary, primarily to handle differences in their output sequence lengths and feature dimensions. To clearly define the operational logic for this alignment, we provide Algorithm 1. This algorithm details the specific steps involved: average pooling to match sequence lengths, followed by a projection layer to align feature dimensions.

## D EXPERIMENT DETAILS

### D.1 DATA

**Pre-training Datasets.** Our pre-training utilized the human reference genomeSchneider et al. (2017) as the primary data source. The genome was segmented and extended to a maximum length of $1,048,576$ base pairs, resulting in a training split of $34,021$ segments, which collectively amount to approximately 35 billion nucleotide tokensNguyen et al. (2023).

**Fine-tuning Benchmarks.** For the fine-tuning benchmarks, we collected data from two primary sources: the Nucleotide Transformer benchmarks (available at `https://huggingface.`

`co/datasets/InstaDeepAI/nucleotide_transformer_downstream_tasks`) and the Genomic Benchmarks (available at `https://github.com/ML-Bioinfo-CEITEC/genomic_benchmarks`). The NT benchmark datasets, originating from five peer-reviewed genomics studies, encompass 18 downstream tasks including promoter prediction, enhancer identification (binary and multi-class), splice site detection, and epigenetic mark classification in yeast. The Genomic Benchmarks provides a curated collection of sequence classification datasets focusing on regulatory elements (such as promoters, enhancers, and open chromatin regions) from human, mouse, and roundworm, derived from both existing literature and novel data mining of public databases.

## D.2 TRAINING DETAILS

**Pra-training.** For the pre-training hyperparameters, we focused on settings suitable for downstream tasks typically involving sequences shorter than 1k base pairs, such as those in the Nucleotide Transformer Benchmarks and Genomic benchmarks. Consequently, we selected a sequence length of 1026 for our pre-training, a value divisible by $k = 6$, which aids in managing the tokenizer mismatch between our character-level student model and the $k$-mer based teacher model. We employed a batch size of 1024. The learning rate was set to $3e - 3$, utilizing the "*cosine warmup*" scheduling strategy. For optimization method, we used "*AdamW*" with $\beta_1 = 0.9$ and $\beta_2 = 0.95$. The pre-training was conducted for 10,000 global steps. Regarding the distillation process, we intuitively chose a $1 : 1$ ratio for balancing the distillation and reconstruction objectives. All pre-training experiments were conducted on a system equipped with $8$ NVIDIA A800-40G GPUs.

**Fine-tuning.** We used the weights of the student model extracted from the pre-training phase, excluding the decoder and other parts of the model. Specifically, we trained on the Nucleotide Transformer Benchmark for 20 epochs and on the Genomic Benchmark for 10 epochs, employing an early stopping strategy. The learning rates and batch sizes for the different tasks are listed in the Table 6 and Table 7. All fine-tuning experiments utilized the same system equipped with eight NVIDIA A800-40G GPUs.

Table 6: HAD Hyperparameter Selection for Nucleotide Transformer Benchmarks.

| Categories | Tasks | LR | Batch Size |
|---|---|---|---|
| Histone markers | H3 | $2e^{-3}$ | 256 |
| | H3K14ac | $2e^{-3}$ | 512 |
| | H3K36me3 | $2e^{-3}$ | 512 |
| | H3K4me1 | $1e^{-3}$ | 256 |
| | H3K4me2 | $1e^{-3}$ | 256 |
| | H3K4me3 | $2e^{-3}$ | 512 |
| | H3K79me3 | $1e^{-3}$ | 256 |
| | H3K9ac | $2e^{-3}$ | 256 |
| | H4 | $1e^{-3}$ | 512 |
| | H4ac | $2e^{-3}$ | 256 |
| Enhancer annotation | Enhancers | $2e^{-3}$ | 512 |
| | Enhancers types | $2e^{-3}$ | 512 |
| Promoter annotation | Promoter all | $1e^{-3}$ | 256 |
| | Promoter no TATA | $1e^{-3}$ | 256 |
| | Promoter TATA | $2e^{-3}$ | 512 |
| Splice site annotation | Splice sites acceptors | $2e^{-3}$ | 256 |
| | Splice sites all | $2e^{-3}$ | 256 |
| | Splice sites donors | $2e^{-3}$ | 256 |

Table 7: HAD Hyperparameter Selection for Genomic Benchmarks.

| Tasks | LR | Batch Size |
|---|---|---|
| Mouse Enhancers | $5e^{-4}$ | 32 |
| Coding vs. Intergenomic | $2e^{-3}$ | 128 |
| Human vs. Worm | $3e^{-3}$ | 256 |
| Human Enhancer Cohn | $3e^{-3}$ | 512 |
| Human Enhancer Ensembl | $1e^{-3}$ | 256 |
| Human Regulatory | $1e^{-3}$ | 256 |
| Human OCR Ensembl | $3e^{-3}$ | 256 |
| Human NonTATA Promoters | $2e^{-3}$ | 256 |

**T-SNE Visualization Details.** To assess the model's ability to generalize across various genomic features, we conducted a t-SNE analysis of model embedding. In addition to the visualizations, we quantitatively evaluated the representation quality using KL Divergence and k-NN accuracy. These metrics provide a quantitative measure of representation quality, beyond fine-tuning benchmarks. KL Divergence measures the discrepancy between high-dimensional and low-dimensional probability distributions, where lower values indicate better preservation of structure. k-NN Accuracy evaluates how well clusters are separated in the low-dimensional space, with higher values indicating better clustering.

We evaluated the model's generalization on key genomic tasks including splice sites, promoters, histone markers, and enhancers. The results showed that our HAD significantly outperforms the NTv2-500M teacher model in both KL Divergence and k-NN accuracy, indicating superior learned representations.

| Combination | Metric | HAD w/o Visible Distillation | NTv2-500M | HAD |
|---|---|---|---|---|
| Enhancer versus Splice sites all | KL Divergence | 2.520 | 1.957 | 0.991 |
| | k-NN Accuracy | 0.193 | 0.260 | 0.668 |
| Enhancer versus H3 | KL Divergence | 2.426 | 1.816 | 0.948 |
| | k-NN Accuracy | 0.221 | 0.299 | 0.665 |
| Promoter all versus H3 | KL Divergence | 1.958 | 1.743 | 0.908 |
| | k-NN Accuracy | 0.232 | 0.287 | 0.700 |
| Promoter all versus Enhancers | KL Divergence | 1.950 | 1.929 | 1.076 |
| | k-NN Accuracy | 0.236 | 0.274 | 0.634 |
| Promoter all versus Splice sites all | KL Divergence | 2.003 | 1.929 | 0.983 |
| | k-NN Accuracy | 0.230 | 0.274 | 0.695 |
| H3 versus Splice sites all | KL Divergence | 2.003 | 1.806 | 0.811 |
| | k-NN Accuracy | 0.230 | 0.284 | 0.719 |

Table 8: Zero-shot evaluation of model representation quality using KL Divergence and k-NN Accuracy across multiple genomic task combinations. Lower KL Divergence and higher k-NN Accuracy indicate better preservation and clustering of learned representations.

### D.3 METRICS

This section defines the evaluation metrics used across Nucleotide Transformer Benchmarks. The specific metric employed varies by task: Matthews Correlation Coefficient (MCC) is used for histone markers and enhancer annotation; F1-score for promoter annotation and splice site acceptor/donor tasks; and accuracy for the "splice site all" task. In these definitions, $TP$ represents True Positives

(correctly identified positive samples), $TN$ represents True Negatives (correctly identified negative samples), $FP$ represents False Positives (negative samples incorrectly identified as positive), and $FN$ represents False Negatives (positive samples incorrectly identified as negative).

**Matthews Correlation Coefficient (MCC).**   The Matthews Correlation Coefficient is a robust statistical rate which produces a high score only if the prediction obtained good results in all four confusion matrix categories (true positives, false negatives, true negatives, and false positives). It is particularly useful when the classes are of very different sizes, offering a balanced measure of classification quality.

$$\text{MCC} = \frac{TP \times TN - FP \times FN}{\sqrt{(TP+FP)(TP+FN)(TN+FP)(TN+FN)}} \tag{9}$$

**F1 Score**   The F1 score is the harmonic mean of precision and recall, providing a single score that balances both metrics. It is often used in scenarios where both false positives and false negatives are important to consider, and is particularly valuable for datasets with imbalanced class distributions.

$$\text{(per-class) F1} = 2 \times \frac{\text{Precision} \times \text{Recall}}{\text{Precision} + \text{Recall}}, \tag{10}$$

where:

$$\text{Precision} = \frac{TP}{TP + FP}$$
$$\text{Recall} = \frac{TP}{TP + FN}$$

**Accuracy**   Accuracy measures the proportion of all predictions that are correct, considering both true positives and true negatives. While it is a straightforward and intuitive metric, accuracy can be misleading on imbalanced datasets where one class significantly outnumbers the others, as a high accuracy might primarily reflect the correct classification of the majority class.

$$\text{Accuracy} = \frac{TP + TN}{TP + TN + FP + FN} \tag{11}$$

## E   BORDER IMPACT STATEMENTS

Our work delves deeper into the potential of knowledge distillation when applied to hybrid architectures in the DNA masked modeling task. This exploration not only advances the representation learning capabilities of DNA foundation models but also broadens the scope of knowledge distillation applications in this specialized domain. By enhancing the representation learning of DNA foundation models, our approach equips these models with a more refined ability to understand and process genomic data. This, in turn, can drive progress in various genomic research and application areas. Furthermore, the extension of knowledge distillation to DNA masked modeling tasks provides a new avenue for improving model performance. Our work paves the way for more efficient and effective DNA modeling approaches, potentially benefiting both research and practical applications in the field of genomics.

## F   ASSETS

We list the existing assets and corresponding licenses in Tab.9.

## G   THE CLARIFICATION OF LLMS USAGE

The methods and experiments presented in this paper do not involve the use of large language models (LLMs). LLMs were solely utilized for enhancing the clarity of the writing and for grammar corrections.

Table 9: Assets and their Licenses

| Asset | License |
|---|---|
| GRCh38Schneider et al. (2017) | CC BY 4.0 |
| Genomic BenchmarksGrešová et al. (2023) | Apache-2.0 |
| Nucleotide TransformerDalla-Torre et al. (2024) | CC BY-NC-SA 4.0 |
| Flash Linear AttentionYang & Zhang (2024) | MIT |
| DNABERTJi et al. (2021) | Apache-2.0 |
| DNABERT2Zhou et al. (2023) | Apache-2.0 |
| HyenaDNA Nguyen et al. (2023) | Apache-2.0 |
| FlashAttention Dao et al. (2022); Dao (2024) | BSD-3-Clause |
| Pytorch Ansel et al. (2024) | BSD-3-Clause |
| Pytorch Lightning Falcon & The PyTorch Lightning team (2019) | Apache-2.0 |
| Huggingface Wolf et al. (2020) | Apache-2.0 |
| Scikit-Learn Pedregosa et al. (2011) | BSD-3-Clause |
| Numpy Harris et al. (2020) | BSD-3-Clause |
| Matplotlib Hunter (2007) | Matplotlib License |

