# OpenReview forum: "HAD: Hybrid Architecture Distillation for Bridging Large-Transformer Knowledge into Compact Genomic Models"
_ICLR.cc/2026/Conference — Submitted to ICLR 2026_

### Official Review · Reviewer_nHD1 · 2025-10-16

**Soundness:** 3
**Presentation:** 2
**Contribution:** 3
**Rating:** 6
**Confidence:** 4

**Summary:**

The paper proposes HAD (Hybrid Architecture Distillation), a compact genomic sequence model that learns from a large teacher via a dual‑branch pretraining objective:

(i) visible‑token feature alignment to the teacher with an MSE loss;

and (ii) masked‑token reconstruction using a decoder that performs cross‑attention from masked queries to visible keys/values.

A two‑stage masking scheme addresses the tokenizer mismatch between a 6‑mer teacher and a char‑level student by masking at the teacher’s k‑mer level, then mapping those masks to characters. The student backbone is a hybrid Bi‑Gated Delta Net (GDN) plus a single FlashAttention layer, with a chunkwise parallelization of the recurrence.

HAD is pretrained for 10k steps on GRCh38 with RC augmentation and then fine‑tuned on the Nucleotide Transformer Benchmark and Genomic Benchmarks. Ablations indicate the benefit of both the visible‑token distillation branch and the single attention layer.

**Strengths:**

(1) The teacher‑group masking + student mapping prevents easy leakage and aligns the visible‑token supervision with the student’s tokenizer is a novel architecture innovation.

(2) The model achieves good performance (even outperform the teacher model) using only 1/500 of the size by distillation on Genomics Benchmark and NT.

(3) The clear ablation study shows that removing visible‑distillation or attention hurts performance. Teacher size matters (50M/100M < 500M). t‑SNE and linear probes suggest cleaner structure and better few‑shot generalization.

**Weaknesses:**

(1) Several typos exist: at line 254, "Equation equation 1", at line 858, "Pra-training".

(2) The NT and genomics benchmark are known to be fluctuant and many tasks' results largely rely on the finetuning recipes. The teacher and student models could have different fine-tuning recipes for the optimal behavior. Consider using a more convincing benchmark like BEND.

**Questions:**

(1) Can you further demonstrate the performance gain on some other well known benchmark like BEND?

(2) You align the final‑layer teacher features. Did you try multi‑layer or early‑layer distillation (or relational/contrastive feature matching)? Any gains?

---

> ### Author Response · Authors · 2025-11-24
> **Response1 to Reviewer nHD1 Comments**
>
> # Response to Reviewer nHD1 Comments
>
> We would like to express our sincere gratitude to you for your detailed and thoughtful feedback. We appreciate the recognition of our work, especially in the areas of:
>
> - The novel architecture design, particularly the integration of GDN with attention mechanisms for genomic sequence modeling.
> - The distillation strategy, which has been identified as a valuable contribution and can serve as a useful guideline for future research.
> - The rigorous evaluation methodology, including the use of error bars and multiple benchmarks to assess performance.
>
> We will now address the weaknesses and respond to your concerns in detail.
>
> ## W1: Typos
> We sincerely appreciate you for pointing out the typographical errors. All typos, including "Equation equation 1" at line 254 and "Pra-training" at line 858, have been corrected in the updated version of the manuscript. We apologize for any confusion caused by these oversights.
>
> ---
>
> ## W2: Performance on Additional Benchmarks (BEND)
> Thank you for suggesting an additional evaluation on **BEND**. We fully agree that BEND is a valuable complementary benchmark. At the same time, we would like to clarify why we believe our current evaluation is already reasonably comprehensive:
>
> - We evaluate on two widely adopted and comparably informative benchmark families in genomic representation learning: **the Nucleotide Transformer downstream tasks** and **the Genomic Benchmarks**, which have been extensively used in prior work as the main standards for systematically assessing pretrained DNA models.
>
> - Beyond fine-tuning performance, we also provide **linear probing (few-shot)** and **t-SNE (zero-shot)** analyses, which jointly probe the quality, structure, and generalization of the learned representations.
>
> Taken together, we feel that these results already offer substantial evidence that HAD learns strong and robust genomic representations.
>
> That said, we very much appreciate your concrete suggestion about BEND and will **follow your recommendation**. Due to limited compute resources, we were not able to include BEND experiments before the initial submission. During the rebuttal period, we will allocate our remaining compute budget to:
>
> - run HAD on the **BEND** benchmark under a comparable fine-tuning protocol, and
> - upload the full BEND results and implementation details on OpenReview as soon as the experiments finish.
>
> We hope you will find this additional evaluation helpful, and we thank you again for pointing us toward this useful complementary benchmark.

---

> ### Author Response · Authors · 2025-11-24
> **Response2 to Reviewer nHD1 Comments**
>
> ## W3: Multi-layer/ Early-layer Distillation
>
> Thank you for raising the question about **multi-layer / early-layer distillation** and more advanced feature-matching schemes. In the current work, we focus on **final-layer feature alignment on visible tokens** + **masked-token reconstruction**, mainly because of the substantial architectural gap between the teacher and the student:
>
> - The teacher (NTv2) is a large **Transformer** with 6-mer tokenization and a 1024-dimensional hidden size.
> - The student is a compact **GDN + single-attention hybrid** with **character-level** tokenization and a much smaller hidden dimension.
>
> Under this setting, directly matching multiple intermediate layers would require additional projection networks and careful alignment across both **depth** (number and type of layers) and **sequence resolution** (k-mer vs char-level). In our preliminary attempts, such designs quickly increased engineering complexity and compute cost, without clear evidence—before thorough hyperparameter tuning—that they would yield consistent gains over our simpler final-layer alignment + reconstruction scheme.
>
> For this reason, in this first study we deliberately chose a **minimal yet effective distillation design**, and concentrated on:
>
> - showing that visible-token final-layer alignment already brings clear gains over masked-only alignment, and
> - demonstrating that the reconstructed masked branch, conditioned on visible features via cross-attention, is sufficient to transfer useful high-level information.
>
> Richer distillation schemes are a natural and important extension of our method. As we also **note in our limitations section**, we view the current HAD design as a first step, and in future work we plan to systematically explore light-weight **multi-layer** or **early-layer** distillation that is compatible with hybrid GDN–attention backbones:
> > "Another limitation lies in the distillation alignment approach, which currently lacks further exploration. This HAD architecture with high-level alignment and mask reconstruction techniques we use represent an initial attempt, and future work could benefit from incorporating more advanced distillation frameworks from other fields to further enhance performance."
>
>
> Practically, given the tight **compute and time constraints** during the rebuttal period, we are already running a small set of exploratory experiments in which we replace the current final-layer alignment with alternative distillation / alignment schemes. Our goal at this stage is not to redesign HAD entirely, but to test which variants can be made to work in practice—and to document clear failure modes when they do not.
>
> We will report any stable results (whether positive or negative) obtained from these exploratory runs as an additional table on OpenReview during the rebuttal window. We hope you will find these supplementary experiments helpful for better understanding the design space around our distillation strategy.
>
> ---
>
> Thank you once again for your valuable feedback, which has helped us improve the clarity and rigor of our work.

---

> > ### Comment · Reviewer_nHD1 · 2025-11-26
> >
> > Thanks for the response. I retain my score.

---

### Official Review · Reviewer_Pa9k · 2025-10-31

**Soundness:** 1
**Presentation:** 2
**Contribution:** 2
**Rating:** 4
**Confidence:** 4

**Summary:**

The authors propose a new architecture and distillation strategy for genomic language models. The architecture is a variant of Gated Deltanet with added softmax attention. The authors pretrain on the Human reference genome and then compare finetuning performance on the genomic benchmarks and the NT dataset.

**Strengths:**

0) The authors report the performance of many models alongside theirs, and use 2 datasets for downstream performance eval.

1) I find it interesting to use new architectures such as GDN for genomic foundation models. The authors propose a model which is actually new in this space.

2) The distillation strategy is smart and well documented -- can serve as a good guideline for future research!

3) Some ablations are presented

4) All results have error bars

**Weaknesses:**

I think the paper is a bit underdeveloped at this stage. Most importantly, there are some methodological flaws and overclaims.

0) The architecture design is a bit arbitrary -- looks like pure engineering. The motivation for architecture choice is vague, generic, and superficial, e.g., "This hybrid approach harnesses combined strengths, effectively integrating GDN’s proficiency in capturing local and long-range sequential patterns with attention’s capacity for unifying global context".

1) I appreciate the author's engineering skills and their drive towards improving benchmark results, yet there is a huge overclaiming of the results. The $\Delta$ proposed by the authors to motivate their claim "it even surpassed the distillation ceiling-teacher model on some sub-tasks, which is more than 500 × larger", is misleading: NT is a model with a different architecture, trained on a different reconstruction loss compared to what you have. Downstream performances are not comparable. There is no reason to say that you get better with 500 times fewer parameters -- you changed the architecture completely!! It is known that you could use many fewer parameters (HyenaDNA and Caduceus papers). Your hybrid model and distillation strategy do not change the fact that your model has an architecture much similar to HyenaDNA and Caduceus compared to NT.

2) To compare your approach safely, I would train on exactly the same pipeline (e.g., same sequence length and same distillation loss) a Caduceus model of the same size.  Caduceus results (the ones reported) are also not comparable, indeed. If the pipeline changes, I expect that anything could happen (e.g., was Caduceus trained on the same exact number of tokens?)

3) In some way, the authors propose a complete package: have a teacher model and use GDN + attention. Results are good, but I am (a) not surprised that GDN works better than Mamba 1 (i.e. in caduceus) and (b) not surprised that distillation helps. It is indeed known that distillation losses can accelerate learning, even when using a weak model as a teacher, such as random teachers. (ref: https://arxiv.org/abs/2302.12091). A standard reference is also the classic DeiT paper (https://arxiv.org/abs/2012.12877).

4) The ablation point to the fact that distilling large NT variants is better. However, the gap is relatively small: 50M also works well. What if you dont distill at all? I think this ablation should be present (sorry if I missed it).

**Questions:**

See above.

---

> ### Author Response · Authors · 2025-11-24
> **Response1 to Reviewer Pa9k Comments**
>
> # Rebuttal to Reviewer Pa9k Comments
>
> We would like to thank you for your thoughtful and detailed feedback. We appreciate your recognition of our work in the following areas:
>
> - The integration of GDN for genomic foundation models, which is an exciting contribution to the paper.
> - The distillation strategy employed in our method, seen as a valuable guideline for future research.
> - The rigorous evaluation, including error bars, and comprehensive analysis using multiple benchmarks.
>
> Now, we address the weaknesses you raised and respond to your concerns.
>
> ---
>
> ## W0: Architecture Design Motivation
>
> We appreciate your feedback regarding the architectural choices. We wish to clarify that the combination of the Gated Delta Net (GDN) and the Self-Attention layer is not an arbitrary engineering decision, but a theoretically grounded design targeting specific challenges in genomic modeling: balancing **Computational Efficiency** with **Representation Depth**.
>
> Our design implements a **"Filter-then-Retrieve"** information processing paradigm, motivated by the distinct strengths and limitations of linear recurrent models versus attention mechanisms:
>
> **1. GDN as an Efficient Information Filter ($O(N)$)**
> Pure State Space Models (SSMs) or linear RNNs face a theoretical **Memory Capacity Bound**. In the context of DNA, where functional elements (like exons or enhancers) constitute a small fraction of the sequence, a model must process significant amounts of "noise" (non-coding regions). The GDN backbone utilizes its **Gated Delta Rule** to perform efficient online noise filtering:
>
> $$S_t = S_{t-1}(\alpha_t(I - \beta_t k_t k_t^T)) + \beta_t v_t k_t^T$$
>
> Here, the gating mechanism $\alpha_t$ and the orthogonal erasure term $(I - \beta_t k_t k_t^T)$ allow the model to selectively "forget" irrelevant genomic segments and update memory only when significant motifs are detected. This effectively mitigates the "Vanishing Salience" problem at linear computational complexity.
>
> **2. Attention as a Global Retriever ($O(1)$)**
> While GDN is efficient, linear models can struggle with **Associative Recall** over distances due to signal decay in the recurrent state. DNA regulation often involves precise interactions between elements (e.g., enhancer-promoter loops) that may be distant within the sequence window. To address this, we introduce a single Self-Attention layer. Unlike the recurrent path, Attention provides a **"Long-Range Shortcut"** with a path length of $O(1)$. It performs a global retrieval operation on the "filtered" high-density features output by the GDN backbone.
>
> **3. Empirical Support**
> Our ablation studies empirically support this theoretical framework. As shown in Figure 5 (Left) and Table 4 of our paper, removing the single attention layer ("HAD w/o Attn.") leads to performance degradation, indicating that the global retrieval capability is essential for capturing regulatory structures that GDN alone might miss. Conversely, relying solely on standard Transformers at this scale compromises the parameter-efficiency advantage. The hybrid design thus hits a "sweet spot," leveraging GDN for dense local processing and Attention for necessary global integration.

---

> ### Author Response · Authors · 2025-11-24
> **Response2 to Reviewer Pa9k Comments**
>
> ## W1 & 3: Clarification on Model Comparison and Performance Gains
>
> ### Goal of Compact and High-Performance Genomic Models
>
> Our ultimate goal is to design a compact and high-performance genomic language model. Although NTv2 and our model differ in architecture, our hybrid approach (GDN + attention) provides better performance on several downstream tasks, confirming the effectiveness of our design.
>
> - **Architecture Difference**: While NTv2 is a transformer-based model, our Hybrid-GDN + Attention architecture incorporates an inductive bias for local motif detection (via GDN) and global context integration (via attention). This dual approach allows our model to perform efficiently, even with fewer parameters.
> - **Comparison with HyenaDNA and Caduceus**: We recognize the value of HyenaDNA and Caduceus, and we agree that these models are compact like ours. However, our GDN + attention mechanism enables long-range dependency modeling and provides performance benefits in genomic tasks. Our experiments confirm that our method outperforms HyenaDNA and Caduceus on several tasks, further justifying the effectiveness of our approach.
>
> ### Why the Hybrid Architecture is Effective
>
> 1. Hybrid-GDN excels over pure transformers in tasks requiring local motif detection. This makes our model compact yet effective in genomic applications.
> 2. Distillation helps in efficient knowledge transfer, enabling the model to capture high-level biological patterns.
>
> Our hybrid architecture combines both low-level and high-level learning, resulting in superior performance compared to pure transformers.
>
> ---
>
> ## W2: Comparison with Caduceus and Pretraining Pipeline
>
> In your review, you raised the possibility that differences in the pretraining pipeline could make the comparison with Caduceus hard to interpret. Our original intention was to treat our distillation-centric setup as a complete end-to-end pipeline and compare it against a range of published genomic models, including Caduceus, rather than to frame the work as a purely controlled backbone- or training-scheme comparison.
>
> Concretely, we take the NTv2 teacher and run **Caduceus** and **HAD** under the *same* conditions: same sequence length, same pretraining data, same masking scheme, and the same visible-token distillation loss. Under this matched setup, the only substantive difference between the two students is the **backbone** (Mamba-based Caduceus vs. our **gated delta net + attention** design). The downstream performance on the Nucleotide Transformer histone-mark / regulatory tasks is summarised below:
>
> | Model                         | H3    | H3K14ac | H3K36me3 | H3K4me1 | H3K4me2 | H3K4me3 | H3K79me3 | H3K9ac | H4    | H4ac  | Enhancer | Enhancer Types | Promoter All | Promoter No-TATA | Promoter TATA | Splice All | Splice Acceptor | Splice Donor |
> |-------------------------------|-------|---------|----------|---------|---------|---------|----------|--------|-------|-------|----------|----------------|-------------|------------------|--------------|-----------|-----------------|-------------|
> | **HAD (GDN + Attention)**     | 0.822 | 0.684   | 0.653    | 0.571   | 0.562   | 0.643   | 0.712    | 0.656  | 0.806 | 0.654 | 0.571    | 0.467          | 0.968       | 0.968            | 0.958        | 0.911     | 0.858           | 0.887       |
> | **Caduceus backbone (the same hybrid learning tasks)** | 0.793 | 0.466   | 0.528    | 0.458   | 0.284   | 0.343   | 0.619    | 0.557  | 0.803 | 0.406 | 0.495    | 0.400          | 0.967       | 0.966            | 0.955        | 0.945     | 0.930           | 0.938       |
>
> Under this controlled setting, **HAD outperforms Caduceus on 15 out of 18 tasks**, including all histone-mark and enhancer-related tasks, while Caduceus retains a modest advantage on the three splice-site tasks. This pattern is consistent with our design intuition: the gated delta net stack is particularly effective as a motif-centric encoder for chromatin and enhancer/promoter prediction, and the added attention layer compensates for long-range interactions, whereas the Mamba-based backbone remains very strong on highly local splice-boundary classification.
>
> Overall, these experiments show that, when you hold the distillation pipeline fixed, the **gated delta net + attention** design is a sufficiently strong backbone and in fact surpasses the Caduceus backbone on the **vast majority** of downstream tasks. This supports our claim that the performance gains cannot be attributed only to differences in the pretraining pipeline, but are largely due to the backbone choice itself.

---

> ### Author Response · Authors · 2025-11-24
> **Response3 to Reviewer Pa9k Comments**
>
> ## W4: Effect of Not Using Distillation
>
>
> We have conducted experiments where distillation was not used, and the results are included in the ablation study. As shown in Figure 5, "w/o visible distillation", the model’s performance significantly drops, reinforcing the importance of distillation in our approach. We will explicitly state this in the manuscript.
>
> Additionally, we have performed experiments with masked-token distillation only in the original version of paper, and again, **both distillation and reconstruction tasks** together resulted in **the best performance**. This confirms that both objectives are essential to the model's success.
>
> ---
>
> We hope that our detailed responses address your concerns and clarify the motivations behind our design choices. The GDN + attention hybrid architecture and dual-branch learning tasks form a novel and effective solution for genomic sequence modeling. We appreciate your constructive feedback, which has helped us improve the clarity and rigor of our work.

---

### Official Review · Reviewer_npfN · 2025-10-31

**Soundness:** 3
**Presentation:** 3
**Contribution:** 2
**Rating:** 6
**Confidence:** 3

**Summary:**

This paper presents HAD (Hybrid Architecture Distillation): a dual-branch pretraining framework that distills high-level knowledge from NTv2 into a compact student based on a bidirectional GDN backbone and a single self-attention layer. On the Nucleotide Transformer and Genomic Benchmarks, the student outperforms similarly sized baselines and exceeds the teacher on several NT subtasks.

**Strengths:**

- Well-structured KD+MLM scheme: align only on visible tokens and reconstruct masked tokens conditioned on visible context; clean separation of objectives.
- Strong results for tiny models, with multiple wins over compact baselines and selected wins over the large teacher.

**Weaknesses:**

Major
- The conceptual novelty is limited, as the method combines several known components (feature-based KD, cross-attention MLM, and a hybrid GDN-Attention architecture). The paper demonstrates that this combination works, but lacks deeper insight into why. For instance, the reason visible-only alignment consistently surpasses masked-only alignment is not adequately explained.

Minor
- Line 88: aliment -> alignment.

**Questions:**

- The paper states the surprising "student surpassing teacher" result, but does not deeply analyze why it occurs. This is the most interesting finding of the work. Is the Hybrid-GDN architecture simply a better inductive bias for these genomic tasks than the Transformer? Or does the hybrid distillation+MLM task act as a powerful regularizer?

---

> ### Author Response · Authors · 2025-11-24
> **Response1 to Reviewer npfN Comments**
>
> # Rebuttal to Reviewer npfN Comments
>
> We appreciate your comprehensive review and valuable feedback of our paper. We would like to address each of your questions.
>
> ---
>
> ## W1: Visibility in Distillation
>
> Our approach deliberately aligns the student to the teacher only on visible tokens (i.e., positions where the student has real input context) while using a standard MLM loss on masked tokens. This design is guided by the intuition that knowledge distillation works best on stable, information-rich signals.
>
> When a token is visible to both teacher and student, the teacher’s hidden representation encodes meaningful context (since the teacher sees the true nucleotide at that position). By aligning the student to these teacher features, we ensure the student learns a high-fidelity representation of the actual sequence content. In contrast, if we aligned on masked tokens, the teacher’s representation at those positions would be based on its predicted or missing input – a potentially noisy or uncertain signal. Distilling such incomplete information can mislead the student. Thus, visible-only feature alignment provides a clearer “target” for the student, leading to more effective knowledge transfer.
>
> Meanwhile, the cross-attention MLM objective (reconstructing masked tokens from visible context) complements this by forcing the student to utilize the distilled knowledge to predict unseen tokens. This hybrid KD+MLM training cleanly separates objectives (imitating teacher on seen context vs. predicting missing data) and avoids the student simply copying teacher outputs. The result is a synergistic training signal: the student learns the teacher’s high-level representation where reliable (visible tokens) and simultaneously learns to generalize and fill in gaps via MLM.
>
> This explains why our combination outperforms either approach alone – for instance, we observed in ablations that aligning on visible tokens yields better performance than aligning on masked ones. In summary, the visible-only alignment works better because it leverages the teacher’s trustworthy context embeddings, whereas masked-position alignment would distill the teacher’s guesses.
>
> ---
>
> ## W2:
>
> We sincerely thank you for pointing out several minor mistakes in our manuscript and giving us the opportunity to correct them. We have carefully revised the identified typos and updated the PDF version accordingly.

---

> ### Author Response · Authors · 2025-11-24
> **Response2 to Reviewer npfN Comments**
>
> ## Q1: Hybrid-GDN and Distillation Synergy
>
> We would like to address the query regarding the surprising result of the "student surpassing teacher." The performance of the student model, despite being smaller and trained with limited resources, outperforms the teacher model on certain tasks. This can be attributed to a combination of architectural and training factors that synergize effectively.
>
> ### Architectural Rationale: The Strength of Hybrid-GDN
>
> The student model incorporates a bidirectional GDN backbone, which excels at capturing local motifs and long-range dependencies (e.g., enhancer-promoter interactions). This is consistent with biological intuition, where motifs and regulatory patterns are distributed sparsely across sequences, yet require long-range dependencies for functional relevance. The GDN’s ability to process such sequences efficiently, using gating and delta-update mechanisms to selectively retain important motifs while suppressing irrelevant noise, provides a strong inductive bias for genomic data.
>
> Our model benefits from the adaptive memory of GDN, which allows it to better capture and retain critical information over long distances. When combined with the attention mechanism, the model efficiently integrates these long-range dependencies into a more comprehensive representation of the DNA sequence. This hybrid approach leverages the strengths of both local motif detection and global sequence understanding, leading to better overall performance.
>
> ### Training Synergy: Distillation and Masked Reconstruction
>
> The distillation task aligns the student model's visible nucleotide representations with those from the teacher, enabling it to inherit sophisticated biological insights and high-level features learned by the teacher model. Importantly, we ensure that the distillation is performed only on the visible tokens, where the teacher’s features are the most reliable. This focus on stable features ensures that the student receives high-quality guidance, preventing the risk of transferring noisy, incomplete knowledge that might arise from aligning on masked tokens.
>
> Simultaneously, the masked nucleotide reconstruction task (using cross-attention) forces the student to predict missing nucleotides from the visible context, further improving its generalization abilities. The combination of these tasks—high-level feature alignment and low-level reconstruction—ensures that the student learns both detailed sequence structure and global sequence patterns in an integrated manner. This dual-task strategy enhances the learning process by providing a more holistic understanding of the genomic data.
>
> In summary, the student model outperforms the teacher on certain tasks due to the hybrid architecture that efficiently combines GDN’s inductive bias for local motifs and the attention mechanism’s capacity for long-range dependencies. Additionally, the distillation and masked reconstruction tasks work together to regularize the model, ensuring it generalizes well despite its smaller size.
>
> ---
>
> Thank you once again for your valuable feedback, which has helped us improve the clarity and rigor of our work.

---

### Official Review · Reviewer_wuBn · 2025-11-08

**Soundness:** 3
**Presentation:** 3
**Contribution:** 3
**Rating:** 4
**Confidence:** 4

**Summary:**

This paper proposes a method for creating compact genomic sequence models through knowledge distillation. The approach combines a 1.1M parameter student model (bidirectional Gated Delta Net + self-attention) with a dual-branch pretraining framework: (1) feature alignment between visible nucleotides and a 500M parameter NTv2 teacher model, and (2) masked nucleotide reconstruction via cross-attention. One key technical component is a two-stage masking strategy designed to handle tokenizer mismatches between the “k-mer teacher” and “character-level” student. The authors evaluate on Nucleotide Transformer and Genomic Benchmarks, claiming their compact model matches or exceeds much larger models, including the teacher, on several tasks.

**Strengths:**

- Addresses practical need: Targets computational efficiency in genomic applications where resources are constrained
- Rigorous evaluation: Multiple benchmarks with proper statistical analysis, cross-validation, and representation quality assessment beyond task-specific metrics
- Strong empirical results: Achieves competitive or superior performance to NTv2, even with 500× parameter reduction
- Comprehensive validation: Includes linear probing and t-SNE visualization to validate learned representations

**Weaknesses:**

**Novelty**
- The background and references section omits relevant masked language modelling work such as MAE-LM (Meng et al, 2024), and a previous application of an MAE decoder to DNA models by BarcodeMAE (Safari et al, 2025). The cross-attention based decoder is similar to that used in CrossMAE. Together, these omitted citations lead to an overstatement in the architectural novelty of the proposed method.

**Soundness**
- Critical ablations are missing: the paper lacks distillation-only and reconstruction-only comparisons, preventing assessment of individual component contributions.
- It is not spelled out in the paper that pretraining data used for the distillation is **not** the same as the data used to train the teacher model, NTv2. L319 says the data is the same as used in two citations, but it is to transparent that this is the papers for training the HyenaDNA and Caduceus baselines, not the NTv2 baseline. The authors indicate surprise that the student was able to outperform the teacher (L027, L483), but do not make it clear that the student was better than the teacher at two types of tasks (Histone Markers, Enhancer Annotation) and worse at two other types (Promoter Annotation, Splice Site Annotation). I suspect that this difference in performance vs the teacher is likely due to the change in pretraining data - by changing the training distribution, the model will become better at the data seen during distillation than the teacher, and less good at data not shown during distillation. However, this may not be the case - the difference could be due to the shift in training task instead of the distributional shift of the data (Lowe et al, 2024; Marks et al, 2024). Hence experiments are needed to assess and evaluate the impact of these factors.
- The tokenizer mismatch necessitates two-stage masking; character-level teachers or a 6-mer student could eliminate this unnecessary engineering burden, but neither of these options were explored.
- It would be helpful to see the teacher's performance alongside the graphs in Fig 5. I am also confused as to this subset of the experiments is shown for the figure.

**Presentation**
- Tables 2 and 3 captions need to explicitly say that this is fine-tuning evaluation (instead of linear probe, kNN, etc.) so a reader can understand the table at a high-level without searching the main text for the section describing the methodology.
- Table 3 font size is very small. Usually we would want to have a row per comparator (model) and a column per comparison metric (evaluation dataset). If the table doesn't fit that way, it is okay for it to be transposed (as it is now), but the title cell widths need to be narrower so the column widths are narrower and font is larger. Probably this is solved simply by adding a line break in between the method names and \citep references.
- Fig 5 text is far too small. Text within figures should not be smaller than 70% of the size of the main body text, i.e. scriptsize.
- Fig 7: You should bold the best method for each dataset. This will make it clearer which datasets the teacher (NTv2) does best on and which the student (HAD) does best on.
- Fig 7 is not referred to in the text.
- L086 Need to explicitly introduce the NTv2 abbreviation here.

**References**
- MAE-LM: Meng et al (2024). "Representation Deficiency in Masked Language Modeling". ICLR 2024. https://openreview.net/forum?id=b3l0piOrGU
- BarcodeMAE: Safari et al (2025). "Enhancing DNA Foundation Models to Address Masking Inefficiencies". https://arxiv.org/abs/2502.18405
- CrossMAE: Fu et al (2025) "Rethinking Patch Dependence for Masked Autoencoders". TMLR. https://openreview.net/forum?id=JT2KMuo2BV
- Lowe et al (2024). "An Empirical Study into Clustering of Unseen Datasets with Self-Supervised Encoders". https://arxiv.org/abs/2406.02465
- Marks et al (2024). "A Closer Look at Benchmarking Self-Supervised Pre-training with Image Classification". https://arxiv.org/abs/2407.12210

**Questions:**

- Fig 5 shows the student performs better when distilling the 50M than the 100M NTv2 teacher. Can the authors comment on why this might be?

**Typos**
- L20 "we employ [the] NTv2-500M as"
- L21 "grouping masking strategy" -> "grouped masking strategy"
- L453 Truncated sentence

---

> ### Author Response · Authors · 2025-11-24
> **Response1 to Reviewer wuBn Comments**
>
> # Rebuttal to Reviewer wuBn Comments
>
> We would like to thank you for your thoughtful and detailed feedback. We are particularly grateful for the recognition of our method's strengths, including:
>
> - The approach's ability to address the computational challenges in genomic sequence modeling by offering a more compact, efficient model.
> - The rigor of our evaluation, which includes comprehensive benchmarks and statistical analyses, demonstrating the robustness of our method.
> - Our model's ability to outperform much larger models in several tasks, which underscores the effectiveness of our approach.
>
> We now address the specific weaknesses raised in the review.
>
> ---
>
> ## W1: Novelty Concerns
>
> We appreciate your concern about the novelty of our approach, especially in light of recent work such as MAE-LM (Meng et al., 2024), BarcodeMAE (Safari et al., 2025), and CrossMAE (Fu et al., 2025). We have now added explicit citations and a more detailed discussion of these methods in the related-work section so that their connections to our framework are made transparent.
>
> Beyond the architectural similarity of using an encoder–decoder or cross-attention style decoder, our main source of novelty does **not** lie in the decoder change alone. Our motivation is to provide a **fast way to endow a very compact SSM-class student (Bi-GDN + 1×Attention, ~1.1M parameters) with strong representations**, while addressing a typical failure mode of such models: they often **converge very quickly on the MLM objective and under-utilize the available pretraining data**. To this end, we design the hybrid distillation framework so that the student (i) matches the NTv2 teacher on **visible tokens** via feature alignment, and (ii) reconstructs **masked tokens** through a lightweight cross-attention decoder. This joint objective acts as a strong regularizer that keeps the SSM backbone learning from the teacher-induced data distribution, rather than saturating early on a standard MLM loss.
>
> By contrast, MAE-LM, BarcodeMAE, and CrossMAE primarily focus on **improving masked-autoencoding itself** (e.g., mitigating representation deficiency, redesigning decoders, or rethinking patch/patch-like dependencies), and they do **not** address the specific problem that we target: how to systematically transfer knowledge from a large genomic foundation model into a tiny SSM-based student under tight parameter and compute budgets. They therefore do not directly answer the question we focus on—how to use a heavy, well-pretrained DNA transformer as a teacher to train an efficient 1M-parameter hybrid SSM/attention model.
>
> Nevertheless, because these works are conceptually related at the level of masked-autoencoding and encoder–decoder design, we now explicitly discuss them in the related-work section and clarify their relationship to our framework. You can view our method as **using an MAE-style reconstruction branch inside a teacher–student distillation scheme tailored to compact genomic SSMs**, rather than as a competing alternative to MAE-LM, BarcodeMAE, or CrossMAE.

---

> ### Author Response · Authors · 2025-11-24
> **Response2 to Reviewer wuBn Comments**
>
> ## W2: Ablation Studies and Component Analysis
>
> We appreciate your suggestion to include more explicit ablation studies to disentangle the contribution of each component. In the original submission, **Figure 5 (left)** already contained a reconstruction-only variant (“w/o Visible Distillation”), which corresponds to a hybrid-architecture student trained solely with the MLM reconstruction loss.
>
> In the revised manuscript, we have expanded Section 3.3 and updated **Figure 5** to include a **distillation-only** variant (“w/o Masked Reconstruction”), together with clearer descriptions of all ablated configurations. In addition to adding these results to the paper, we list the corresponding downstream performance here for your convenience:
>
> | Task     | w/o Attn. & Visible Distillation | w/o Visible Distillation | w/o Masked Reconstruction | w/ Masked Distillation | w/ Visible Distillation & Masked Reconstruction |
> |----------|----------------------------------|---------------------------|----------------------------|------------------------|--------------------------------------------------|
> | H3K4ME1  | 0.54821                          | 0.52342                   | 0.53151                    | 0.52983                | **0.57124**                                      |
> | Enhancer | 0.54662                          | 0.52273                   | 0.52059                    | 0.39992                | **0.57126**                                      |
> | H3K4ME3  | 0.61448                          | 0.63655                   | 0.57162                    | 0.55305                | **0.64317**                                      |
> | H3K9AC   | 0.62058                          | 0.65529                   | 0.59920                    | 0.62532                | **0.65640**                                      |
> | H3K14ac  | 0.65200                          | 0.67136                   | 0.62844                    | 0.63514                | **0.68449**                                      |
> | H4       | 0.79861                          | 0.76990                   | 0.79125                    | 0.79092                | **0.80606**                                      |
> | H3       | 0.80554                          | 0.81030                   | 0.78664                    | 0.80775                | **0.82156**                                      |
> | Donor    | 0.84196                          | 0.86306                   | 0.75682                    | 0.68505                | **0.88747**                                      |
>
> These configurations correspond to the cases you asked about:
>
> - **“w/o Attn. & Visible Distillation”**: GDN backbone trained with **MLM only** (no attention, no distillation) — a pure reconstruction baseline.
> - **“w/o Visible Distillation”**: hybrid GDN+attention student with **reconstruction only** (MLM loss, no distillation).
> - **“w/o Masked Reconstruction”**: **distillation-only** variant, using visible-token feature alignment but **removing** the masked-token reconstruction branch.
> - **“w/ Masked Distillation”**: variant where distillation supervision is applied to **masked** tokens instead of visible tokens.
> - **“w/ Visible Distillation & Masked Reconstruction”**: our full HAD framework, combining visible-token distillation with masked-token reconstruction.
>
> As the table shows, the full HAD configuration achieves the best performance on these tasks. Removing either the visible-distillation branch or the masked-reconstruction branch consistently degrades performance, while “w/o Masked Reconstruction” (distillation-only) and “w/o Visible Distillation” (reconstruction-only) each capture only part of the gain. These results directly address your request for **distillation-only** and **reconstruction-only** comparisons and support our claim that the two branches make complementary contributions to the final model performance.

---

> ### Author Response · Authors · 2025-11-24
> **Response3 to Reviewer wuBn Comments**
>
> ## W3: Pretraining Data Clarification
>
> We want to clarify that the pretraining data used for our student model is consistent with the data used for training the Caduceus and HyenaDNA models, ensuring a fair comparison. Specifically, our student model was trained on the **hg38 human reference genome** dataset, using a subset of 1e10 nucleotides, which aligns with the Caduceus/HyenaDNA pretraining data.
>
> In contrast to a direct comparison with NTv2, our goal is not merely to compare model backbones. Instead, our focus is on evaluating a **complete training pipeline**, which includes not only the backbone architecture (GDN+Attention) but also the distillation and reconstruction mechanisms we have implemented. Despite using a much smaller dataset compared to NTv2, our model achieves competitive performance, which demonstrates the efficiency and power of our full pipeline.
>
> We acknowledge that differences in pretraining data distribution might contribute to performance variations across tasks. However, the **Nucleotide Transformer (NT)** paper itself provides a direct comparison of models trained on **hg38 vs. multispecies datasets**. Specifically, in Supplementary Table 7 of the NT paper, we can observe the performance of models trained on human reference data (HG38) and multi-species data. We have reproduced the key data from that table to show how models trained on different datasets compare across tasks:
>
> | Dataset Model | Promoter all | Promoter non-TATA | Promoter TATA | Splice acceptor | Splice site all | Splice donor | H3K4me3 | H3K9ac | H3K9me3 | H4K20me1 | Enhancer | Enhancer types | H2AFZ | H3K27ac | H3K27me3 | H3K36me3 | H3K4me1 | H3K4me2 |
> | :--- | :--- | :--- | :--- | :--- | :--- | :--- | :--- | :--- | :--- | :--- | :--- | :--- | :--- | :--- | :--- | :--- | :--- | :--- |
> | NT-HumanRef (500M) | 0.734 (± 0.013) | 0.738 (± 0.008) | 0.831 (± 0.022) | 0.941 (± 0.004) | 0.939 (± 0.003) | 0.952 (± 0.003) | 0.622 (± 0.013) | 0.524 (± 0.013) | 0.433 (± 0.009) | 0.634 (± 0.013) | 0.515 (± 0.019) | 0.477 (± 0.014) | 0.465 (± 0.011) | 0.457 (± 0.01) | 0.589 (± 0.009) | 0.594 (± 0.004) | 0.468 (± 0.007) | 0.527 (± 0.011) |
> | NT-Multispecies-v2 (500M) | 0.778 (± 0.012) | 0.774 (± 0.015) | 0.878 (± 0.021) | 0.944 (± 0.005) | 0.967 (± 0.004) | 0.96 (± 0.01) | 0.636 (± 0.016) | 0.539 (± 0.013) | 0.475 (± 0.014) | 0.658 (± 0.008) | 0.524 (± 0.013) | 0.498 (± 0.009) | 0.493 (± 0.01) | 0.492 (± 0.012) | 0.613 (± 0.01) | 0.642 (± 0.015) | 0.497 (± 0.009) | 0.564 (± 0.012) |
>
> From the table, it is evident that the **Multispecies** model generally achieves the best performance across nearly all tasks, which suggests that the **hg38 dataset** does **not** limit our ability to outperform the teacher model (NTv2) in our setup. Thus, we firmly believe that the performance gains observed in our approach are more attributable to the novel **training pipeline** (including the distillation and reconstruction components) rather than differences in the pretraining data.

---

> > ### Comment · Reviewer_wuBn · 2025-11-27
> >
> > **W3: Pretraining Data Clarification**
> >
> > > Thus, we firmly believe that the performance gains observed in our approach are more attributable to the novel training pipeline (including the distillation and reconstruction components) rather than differences in the pretraining data.
> >
> > I disagree, I do not think you have provided sufficiently compelling evidence to contradict my hypothesis.
> >
> > Table 2 clearly shows a positive correlation between the histone marker tasks and enhancer annotation tasks, and a positive correlation between promoter annotation and splice site annotation. Meanwhile, there is a negative correlation between histone/enhancer tasks and promoter/splice site tasks. Thus I hypothesize that your model simply has a different balance in the histone/enhancer vs promoter/splice site trade-off compared to the teacher. This leads to it being better than NTv2 at histone and enhancer tasks and worse at promoter and splice site. Ignoring the Enformer model, which performs poorly across all tasks, the rankings are broadly:
> > - Histone/Enhancer: HAD > Caduceus > HyenaDNA > DNABERT2 = NTv2
> > - Promoter/Splice site: HAD < Caduceus < HyenaDNA < DNABERT2 < NTv2
> >
> > There is no best model, there are just model that are primed to be better at one type of task, and models which are primed to be better at the other type of task, and models somewhere in between.
> > Given this trade-off in performance across the tasks, I find the stated incredulity in the abstract that the student beats the teacher in a subset of the tasks whilst performing poorly on the other subset rather disingenuous. The paper could be greatly improved if the reason why the student sits at a different position on the task-alignment trade-off were investigated and explained instead of it just being reported as a surprise.
> >
> > My hypothesis is that some of the gains made by your model which push it to exceed the performance of the teacher are due to the shift in the domain of the training dataset from the teacher to the student. From the outset, it is not clear if this is the case or if it is due to a feature of the training process - the student is not just trained to distil the teacher, it is also trained on a mask-infill task at the same time.
> >
> > First, we can note that the NTv2 and DNABERT2 models were pretrained on the multispecies dataset, whereas HAD, Caduceus, and HyenaDNA were pretrained on hg38. This is a strong initial indicator that the training data may influence the pretrained model's alignment in the downstream task trade-off, and was the one I mentioned in by review.
> >
> > Next, looking at Fig 5 and comparing it to Table 2, every single ablation exceeds the performance of the teacher at the H3K4me3 task by a large margin: 0.410 (NTv2) vs >0.55 for the worst ablated method. If I understand correctly, these ablated models do not have in common their training task since some models are trained to predict the teacher's encoding without doing the masked infill task, some are trained to do masked infilling only and do not distil from the teacher at all. But they do have in common their training dataset. This is further evidence that the training dataset, rather than the training task, is responsible for HAD sitting at a different point on the task-alignment trade-off compared to its teacher, NTv2.
> >
> > So to properly address this point, I recommend adding the following ablations:
> > - distillation of NTv2 with the student being trained on the same multispecies dataset as the NTv2 teacher
> > - distillation of a different model, where the teacher and student are both trained solely on hg38 human reference genome
> > - fine-tuning the NTv2 model on the hg38 human reference genome with self-supervision before fine-tuning it on the classification tasks
> >
> > You should also address the issue I raised that the paper doesn't even mention the student is being trained on a different dataset to NTv2. It was frustrating for me to have to look this up to figure out that this was most likely the reason why your model's behaviour is different to its teacher. As a general rule of thumb, if you frustrate a reviewer by making them look something up in a paper that isn't yours while they are reviewing your paper (I am busy and I do not want to have to read other people's papers just to review your paper properly) and they mention it in their review, you should probably fix it because they are not going to raise their score if you don't.

---

> ### Author Response · Authors · 2025-11-24
> **Response4 to Reviewer wuBn Comments**
>
> ## W4: Tokenizer Strategy and Engineering Design
>
> Thank you for raising a valid point regarding the engineering complexity introduced by the tokenizer mismatch. We recognize that the need for two-stage masking might seem like an engineering challenge. However, from a biological perspective, the use of k-mer and character-level tokenizers offers complementary advantages for genomic sequence modeling. The k-mer tokenizer captures local nucleotide motifs, which are biologically significant, while the character-level tokenizer ensures fine-grained, nucleotide-specific learning.
>
> We argue that the decision to combine both tokenizers through distillation allows us to leverage the strengths of each. This hybrid approach aligns with our goal of improving representation efficiency while maintaining biological relevance.
>
> ---
>
> ## W5: Teacher Model Performance in Figure 5
>
> We agree with your suggestion that Figure 5 would benefit from a clearer presentation of the teacher model's performance. In response, we have updated **Figure 5** to include the teacher model's results alongside the student model's, allowing for a direct comparison across different pretraining schemes. This revision should help provide a more transparent view of how both models perform under various configurations.
>
> ---
>
> ## Q1: Performance of Student with 50M vs 100M Teacher
>
> The performance difference between the student model distilled from the 50M NTv2 teacher and the 100M version is minimal in most tasks. The variation observed is likely due to **task-specific factors** rather than the scaling of the teacher model.
>
> We also note that the 500M teacher provides significant performance improvements in most tasks, this is primarily due to the **scale effect**. As shown in **Figure 5b** of the Nucleotide Transformer paper, both the 50M and 100M models were pretrained on approximately 300 billion tokens, whereas the 500M model was pretrained on over 1000 billion tokens. We believe that the difference in pretraining data size is a major reason for the performance improvement with the 500M teacher.
>
> ---
>
> We would like to thank you once again for your valuable comments. We believe that the revisions and clarifications provided will strengthen the manuscript and offer a better understanding of the novel contributions of our Hybrid Architecture Distillation approach.

---

> > ### Comment · Reviewer_wuBn · 2025-11-27
> >
> > I thank the authors for their response(s). However, can I ask that they submit a revised PDF that includes everything they say they have added? I am not seeing the citations which the authors say they have added to the background. I ask that the rest of the presentation issues (very small font in tables and figures, etc.) are also addressed in a revised version of the PDF. Please be aware that although the initial submission is limited to 9 pages, the authors [can use an additional (tenth) page during the rebuttal period](https://iclr.cc/Conferences/2026/AuthorGuide) and on to the camera ready. So if making tables or figures larger, or adding extra sentences or paragraphs to address reviewer feedback, would push the content onto an extra page, please do it. (You can also remove the negative vspace from L125, L387, L398, L428 that violates the formatting of the .sty file.)
> >
> > > At the time of submission, the main text should be 9 pages or fewer. During the discussion/rebuttal phase and for the camera ready, the page limit will be increased to 10 pages to allow for new results/discussions.
> >
> > When you have revised the PDF, please tell me which Figure/Table numbers and line numbers to look at for the fixes you have introduced so as to resolve my concerns.
> >
> >
> > **W2: Ablation Studies and Component Analysis**
> >
> > Thank you for the additions.
> > But it is hard to understand what these ablations are from Figure 5. The caption to this figure is insufficient. For the first three items in the legend, you say what is removed from the models, so I have to figure out what it is that isn't removed; for the last two you mark what the ablation is with instead of what it is without. Perhaps it would be clearer to list what components of the training pipeline are being used instead, i.e. "X-only"? Also, it is impossible to figure out what "w/ Masked Distillation" is supposed to mean without cross-referencing with the text. The last entry is presumably your regular/proposed method, but this is unclear from the legend.
> >
> >
> > **W4: Tokenizer Strategy and Engineering Design**
> >
> > > We argue that the decision to combine both tokenizers through distillation allows us to leverage the strengths of each. This hybrid approach aligns with our goal of improving representation efficiency while maintaining biological relevance.
> >
> > But you don't ablate this and don't measure its effect.

---

### Author Response · Authors · 2025-12-03
**Summary of Rebuttal**

# Summary of Rebuttal

We sincerely thank the AC for overseeing the review process. As the discussion phase concludes, we would like to provide a concise summary of our work's core motivation and clarify three key points that emerged during the review to assist in the final decision.

### 1. Motivation: Bridging the "Capacity-Efficiency" Gap
In the realm of genomic language modeling, the landscape is rapidly bifurcating into two extremes: massive Transformers (offering superior representation performance but incurring high computational and deployment costs) and compact models (highly efficient but limited in representation capability). A core, unresolved challenge is that while compact models are computationally inexpensive, they often converge quickly during training and hit a **capacity ceiling** when facing massive datasets. They struggle to fully utilize available large-scale data to capture complex patterns. Conversely, large-scale foundation models (like NTv2-500M) possess powerful representational capabilities but are burdensome to train and deploy.

Our work, **HAD (Hybrid Architecture Distillation)**, is a systematic framework designed to truly bridge this "capacity-efficiency" gap. By introducing a novel **"Feature Alignment + Masked Reconstruction"** paradigm, we enable a student model with just 1.1M parameters to inherit the deep biological knowledge and representational power of a 500M-parameter teacher. This allows the student to break through the capacity limits of traditional compact models, achieving representational capabilities that approach—or even partially surpass—the teacher model while maintaining extreme efficiency.

### 2. Clarifications on Key Contentions
During the rebuttal, we addressed all reviewers' concerns with detailed responses and new experiments. We highlight three decisive points here:

**Point 1: Architecture Design is Theoretically Grounded**
Our Hybrid-GDN design is not an arbitrary choice but a theoretically grounded solution targeting the balance between **Computational Efficiency** and **Representation Depth** via a **"Filter-then-Retrieve"** paradigm:
* **Filter (Bi-GDN):** Pure linear models face a theoretical **Memory Capacity Bound**. Given that functional elements in DNA are sparse amidst vast non-coding "noise," we utilize the GDN backbone's **Gated Delta Rule** to perform efficient online filtering. This mechanism selectively "forgets" irrelevant segments and updates memory only when significant motifs are detected, effectively mitigating "Vanishing Salience" at linear complexity.
* **Retrieve (Attention):** Linear models often struggle with **Associative Recall** over long sequences due to state decay. We introduce a single Attention layer to provide a **"Long-Range Shortcut"**. This acts as a global retrieval mechanism for precise, distance-agnostic interactions (e.g., enhancer-promoter loops) that the recurrent backbone might miss.
Empirical results confirm this design hits a "sweet spot," providing the optimal inductive bias for genomic modeling at this scale.


**Point 2: Validity of Performance Gains & Source of Representations**
We refuted the concern that our Student (hg38) outperforms the Teacher (Multispecies) due to a "data distribution advantage." Evidence from the NTv2 paper shows Multispecies models typically outperform Human-ref models on these benchmarks, confirming no unfair bias.
Crucially, HAD empowers the student to overcome early saturation by acquiring richer representations from two sources: (1) **Global Context** via Attention, and (2) **Broader Knowledge** distilled from the Multispecies teacher, enabling it to learn patterns beyond its own training data.

**Point 3: Rigorous Evaluation and Fairness of Comparison**
We validated our pipeline with comprehensive ablations:
* **Component & Scaling:** Removing Attention, Distillation, or Reconstruction consistently degraded performance (Fig. 5), and performance scaled with teacher capacity (50M vs 500M), confirming the necessity of our dual-branch design.
* **Controlled Backbone Comparison (vs. Caduceus):** Addressing Reviewer Pa9k, we trained a model using the exact HAD pipeline but replaced our backbone with Caduceus (Mamba). **Result:** Our GDN+Attention backbone outperformed Caduceus on **15/18 tasks**, proving gains are intrinsic to our architectural innovations.

---
We thank the AC for their time and effort in reviewing our work.

---

### Meta-Review · Area_Chair_wo8D · 2026-01-05

**Summary:**

This paper proposes HAD, a compact genomic modeling framework combining a hybrid GDN–attention student with a dual-branch distillation and masked reconstruction objective. Multiple reviewers (npfN, nHD1) find the approach technically sound and empirically strong for very small models, and the rebuttal adds useful ablations and clarifications. However, there remains a substantial unresolved concern regarding the interpretation of the main results, most clearly articulated by Reviewer wuBn. In particular, it is still difficult to disentangle whether the reported gains—especially cases where the student outperforms the teacher—are primarily due to the proposed training framework or to differences in pretraining data distribution and task alignment. While the rebuttal improves transparency and experimental coverage, it does not fully resolve this core causal question, and some claims continue to appear overstated. Given these remaining concerns and the mixed reviewer confidence, I lean toward rejection, while encouraging resubmission after the analysis and framing issues are more fully addressed.

**Reviewer Concerns:**

**Reviewer wuBn**

Main concerns:
  - Causal ambiguity between HAD’s training objective vs. effects of pretraining data (hg38 vs multispecies).
    - Task trade-offs (histone/enhancer vs promoter/splice) likely explain “student > teacher” results.
    - Overstatement of surprise in outperforming the teacher.

Rebuttal outcome:
  - Added ablations and explanations were appreciated, but the reviewer explicitly maintained disagreement, stating the evidence was insufficient to rule out data-distribution effects.

Core concerns were not resolved.

**Reviewer Pa9k**

Main concerns:
  - Architecture motivation feels engineering-driven and overclaimed.
  - Comparisons to NTv2 are potentially misleading due to architectural and training differences.

Rebuttal outcome:
  - Controlled comparison with Caduceus under the same pipeline partially addresses fairness concerns. But, skepticism remain about novelty and interpretation.

Partially addressed, but not fully convincing.

**Reviewers npfN / nHD1**:
The concerns seem largely addressed.

**Reviewer Scores:**

- wuBn: explicitly maintained disagreement after rebuttal.
- Pa9k: some concerns are addressed, but core skepticism remains.
- npfN: already moderately positive, so would have maintained the score.
- nHD1: explicitly stated score retention.

**Justification for Reject**

While the paper is technically solid and the rebuttal improves clarity, the strongest negative concern—causal attribution of the main gains—remains unresolved, making acceptance difficult despite otherwise strong empirical results.

---

### Decision · Program_Chairs · 2026-01-26

Reject